# Distinct stem-like cell populations facilitate functional regeneration of the *Cladonema* medusa tentacle

**Sosuke Fujita**[1,2], **Mako Takahashi**[3], **Gaku Kumano**[3], **Erina Kuranaga**[2], **Masayuki Miura**[1], **Yu-ichiro Nakajima**[1,2,4]*

1 Graduate School of Pharmaceutical Sciences, The University of Tokyo, Tokyo, Japan, 2 Graduate School of Life Sciences, Tohoku University, Sendai, Japan, 3 Asamushi Research Center for Marine Biology, Graduate School of Life Sciences, Tohoku University, Aomori, Japan, 4 Frontier Research Institute for Interdisciplinary Sciences, Tohoku University, Sendai, Japan

* nakaji97@g.ecc.u-tokyo.ac.jp

**Data Availability Statement:** All relevant data are within the paper and its Supporting Information files.

## Abstract

Blastema formation is a crucial process that provides a cellular source for regenerating tissues and organs. While bilaterians have diversified blastema formation methods, its mechanisms in non-bilaterians remain poorly understood. Cnidarian jellyfish, or medusae, represent early-branching metazoans that exhibit complex morphology and possess defined appendage structures highlighted by tentacles with stinging cells (nematocytes). Here, we investigate the mechanisms of tentacle regeneration, using the hydrozoan jellyfish *Cladonema pacificum*. We show that proliferative cells accumulate at the tentacle amputation site and form a blastema composed of cells with stem cell morphology. Nucleoside pulse-chase experiments indicate that most repair-specific proliferative cells (RSPCs) in the blastema are distinct from resident stem cells. We further demonstrate that resident stem cells control nematogenesis and tentacle elongation during both homeostasis and regeneration as homeostatic stem cells, while RSPCs preferentially differentiate into epithelial cells in the newly formed tentacle, analogous to lineage-restricted stem/progenitor cells observed in salamander limbs. Taken together, our findings propose a regeneration mechanism that utilizes both resident homeostatic stem cells (RHSCs) and RSPCs, which in conjunction efficiently enable functional appendage regeneration, and provide novel insight into the diversification of blastema formation across animal evolution.

## Introduction

Regeneration, the phenomenon of re-forming missing body parts, is widespread among metazoans. Common regenerative processes include wound closure immediately after injury; formation of the cellular source, or blastema, that reconstitutes the lost tissue; and regrowth of the tissue that integrates different cellular behaviors such as proliferation and differentiation [1]. Among these regenerative responses, blastema formation is a critical step that can distinguish regenerative and non-regenerative systems. Indeed, most mammalian and avian species

**Funding:** This work was supported by JSPS/MEXT KAKENHI (grant numbers JP21H05767 to G.K., JP21H05255 to E.K., JP21H04774, JP23H04766 to M.M., and JP17H06332, JP22H02762, JP23H04696 to Y.N.), JST grant (JPMJCR1852 to E.K. and JPMJSP2114 to S.F.), AMED-Aging (JP21gm5010001 to M.M.), and AMED-PRIME (JP22gm6110025 to Y.N.), and NIBB Collaborative Research Program (23NIBB202 to Y.N.). The funders had no role in study design, data collection and analysis, decision to publish, or preparation of the manuscript.

**Competing interests:** The authors have declared that no competing interests exist.

**Abbreviations:** aPKC, atypical protein kinase C; ASW, artificial sea water; DIG, digoxigenin; dpa, day post-amputation; dpi, days post-irradiation; FISH, fluorescent in situ hybridization; FITC, fluorescein; hpa, hours post amputation; PFA, paraformaldehyde; RHSC, resident homeostatic stem cell; RSPC, repair-specific proliferative cell; RT, room temperature.

do not form blastema upon injury while wound closure occurs normally [2]. Understanding blastema formation mechanisms in highly regenerative animals may therefore help us identify the necessary elements to potentially improve our regenerative abilities.

Blastema can be defined as an undifferentiated cellular mass that contains cells with mitotic capacity and appears after damage such as amputation [3,4]. Accumulating evidence has suggested that methods of blastema formation vary among animals with high regenerative abilities [5]. For example, in planarians, pluripotent stem cells called neoblasts that are distributed throughout the body are recruited to the injury site to produce blastema [6]. Salamanders can regenerate adult limbs upon amputation via blastema formation, but the underlying cellular mechanisms vary among species: in the axolotl, tissue-specific stem cells contribute to blastema, while in the newt, muscle fibers dedifferentiate into proliferative progenitor cells to behave as blastema [7]. During zebrafish caudal fin regeneration, both osteoblast-derived dedifferentiated cells and resident progenitor cells migrate to the wound site to form blastema [8,9]. These studies support the idea that the supply of blastema has diversified across the animal kingdom, whose members utilize resident stem/progenitor cells and/or repair-specific de novo proliferative cells to reconstruct lost body parts. While mechanisms of blastema formation have been studied extensively in a limited number of regenerative animals, little is known about their evolutionary characteristics: Which elements are acquired as lineage-specific novelties and which are widely conserved within highly regenerative species? In particular, the current understanding of blastema formation largely relies on bilaterian models, and thus the mechanisms of blastema formation outside of bilaterians remain poorly understood.

Among various regeneration contexts, appendage regeneration is widely observed in bilaterians (e.g., amphibian limbs and fish fins) and is suitable for understanding the evolution of regenerative processes [3,10]. The bilaterian program of blastema formation during appendage regeneration is associated with repair-specific stem/progenitor cells [7,9,11]. Yet, to elucidate the evolutionary history of blastema formation programs, their mechanisms must be studied in early-branching metazoans in addition to bilaterian models.

Cnidarians (corals, sea anemones, hydroids, and jellyfish) are among the earliest branching metazoans, composed of 2 major groups Anthozoa and Medusozoa, forming a diverse phylum that contains over 10,000 species, and stand at a unique phylogenetic position as the sister group to bilaterians (Fig 1A). While cnidarians display considerably divergent morphologies and life cycles, represented by the polyp and medusa stages; their common traits are a diploblastic radially symmetric body along with tentacles as the well-defined appendages that bear the stinging cells, nematocytes (cnidocytes) [12,13]. Although regenerative potential may vary among the group, most documented cnidarian species are capable of regenerating lost tissues and organs, and some can even regenerate their entire body [14,15].

Among cnidarians, polyp-type animals such as *Hydra*, *Hydractinia*, and *Nematostella* have been utilized as models to understand the mechanism of whole-body regeneration, including patterning, body axis formation, and mechanical responses [16–18]. After mid-gastric bisection in *Hydra*, cell proliferation of the stem cells, or i-cells, around the wound site is accelerated by Wnt3a produced from dying cells to generate blastema [19]. Upon decapitation of the colony polyp *Hydractinia*, i-cells remotely located in the body column migrate to the injury site to form blastema [20]. During head regeneration of the sea anemone *Nematostella*, 2 adult stem-like cell populations, fast-cycling cells in the body wall epithelium and slow-cycling cells in the mesenteries, migrate toward the amputation site to form blastema [21]. These studies suggest that the recruitment of resident stem cells to the injury site is a prerequisite for blastema formation after amputation of the body. However, it remains unclear whether repair-specific proliferative cells (RSPCs) constitute the cellular source of blastema during cnidarian regeneration.

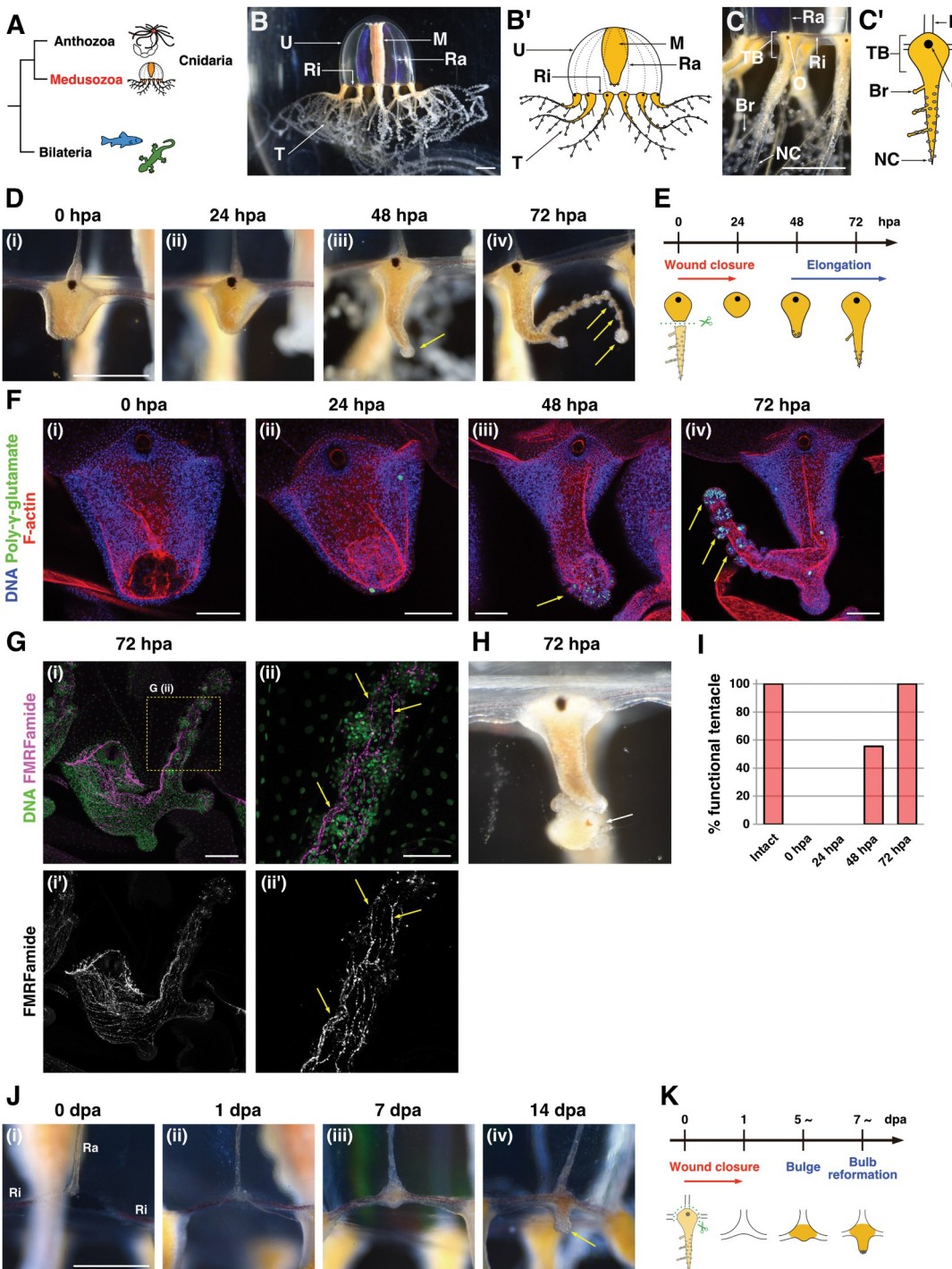

**Fig 1. Regeneration processes and potentials of the *Cladonema* medusa tentacle.** (A) The phylogenetic tree of Bilateria and Cnidaria, composed of Anthozoa and Medusozoa. (B) Medusa of *Cladonema pacificum*. U: umbrella, M: manubrium, Ra: radial canal, Ri: ring canal, T: tentacle. (C) Tentacle of *Cladonema* medusa. O: ocellus, TB: tentacle bulb, Br: branch, NC: nematocyte cluster. (D) Tentacle regeneration processes after amputation while retaining the bulb. hpa: hour post-amputation. Yellow arrows indicate nematocyte clusters. *n* = 234/236 (tentacles). (E) Scheme of the tentacle regeneration that retains the bulb. (F) Distribution of mature nematocytes and localization of muscle fibers during tentacle regeneration. DAPI for poly-γ-glutamate (green) and nuclei (blue), and Phalloidin for F-actin (red). Yellow arrows indicate nematocyte clusters. (G) Neural morphology in the regenerating tentacle stained with the anti-FMRFamide antibody. Yellow arrows indicate neural fibers. (H) The regenerating tentacle is fully functional at 72 hpa. Image of the tentacle capturing prey, brine shrimp (white arrow). (I) The rate of functional tentacles across the regeneration time course. Intact: *n* = 40 (tentacles), 0–72 hpa: *n* = 36. (J) Tentacle regeneration process after

removing the bulb from canals. Yellow arrow in (iv) indicates a nematocyte cluster. Dpa: day post-amputation. $n = 117/219$ (tentacles). Ra: radial canal, Ri: ring canal. (K) Scheme of tentacle regeneration without bulb. The numerical values that were used to generate the graphs in (**I**) can be found in S1 Data. Scale bars: (B–D, J) 1 mm, (F, Gi) 100 μm, (Gii) 50 μm.

In contrast to the sessile polyp stage, medusae, commonly called jellyfish, exhibit a more complex body structure that includes multiple types of muscle such as smooth and striated muscles, developed neural networks, and distinct organs and appendages, which together enable free swimming and avoidance behaviors [22,23]. While most sexually reproducing medusae do not clonally propagate like polyps, medusae can still regenerate various organs and appendages (e.g., manubrium, tentacle, umbrella, gonad, eye) after injury and reconstitute de novo structures after their removal [24–30]. A recent report using the hydrozoan jellyfish *Clytia* has suggested that, upon removal of the entire manubrium, i-cells and differentiated cells localized in neighboring organs migrate to the damage site through the canals, contributing to de novo manubrium regeneration [27]. Despite the importance of proliferating blastema cells in highly regenerative animals, the detailed mechanisms of organ and appendage regeneration in medusae, particularly the mechanism of blastema formation and its specific role are largely unknown.

The hydrozoan jellyfish *Cladonema pacificum* is an emerging jellyfish model that has been utilized to study development and physiology [31–33]. Because *Cladonema* allows for easy lab maintenance with its high spawning rate, it enables monitoring all life cycle stages and exploring organismal responses to different stimuli. The genus *Cladonema* is morphologically characterized by branched tentacles at the medusa stage (Fig 1B) [31,34–36]. The *Cladonema* medusa tentacle is primarily composed of bilayered epithelial tissues (epidermis and gastrodermis) that include muscle fiber, also called epitheliomuscular cells [23]: i-cells, neurons, and nematocytes are located in the epidermal layer while neurons and gland cells are located in the gastrodermal layer. Resident stem cells, i-cells, are localized at the basal side of the tentacle (tentacle bulb), and are thought to give rise to progenitors and differentiated cells during normal development and homeostasis [24,37]. The continuous growth and branching potential of the medusa tentacle makes it an attractive model to understand the mechanisms of appendage morphogenesis, growth, and regeneration in cnidarians.

In this study, we investigate the cellular mechanism of appendage regeneration using the *Cladonema* medusa tentacle. Establishing the *Cladonema* tentacle as an organ that can efficiently and functionally regenerate upon amputation, we show that highly proliferative cells accumulate at the injury site within 24 h to behave as blastema. Pulse-chase experiments using nucleoside analogs as well as dye labeling reveal that most blastema cells are not derived from resident stem cells but rather appear locally after damage. We further identify the role of blastema cells as the principal cellular source of the epithelium in the newly formed tentacle, while resident stem cells contribute to nematogenesis and tissue elongation during both homeostasis and regeneration. These results suggest the existence of 2 distinct proliferative cell populations: blastema as RSPCs and resident stem cells as homeostatic stem cells, both of which collectively enable functional tentacle regeneration. In a broader context, our findings highlight the diversification of blastema formation in non-bilaterian systems, providing an evolutionary insight into animal regeneration.

## Materials and methods

### Animal cultures and surgical manipulations

*Cladonema pacificum* (female strain 6W) medusae were used for this research. The medusae were maintained in plastic containers (V-type container, V-7, V-8, and V-9; AS ONE) at 22°C

in artificial sea water (ASW). ASW was prepared using SEA LIFE (Marin Tech) dissolved in tap water with chlorine neutralizer (Coroline off; GEX Co.) (24 p.p.t). The medusae were fed Vietnamese brine shrimp (A&A Marine LLC).

Before surgical manipulation, the medusae were anesthetized with 7% $MgCl_2$ in deionized water for 2 min in a petri dish (Falcon, 351008). Surgical manipulations were performed with micro scissors (Teraoka, NY33408). After manipulations, the medusae were quickly returned to ASW. When the tentacle was amputated with the bulb left intact, the medusae were fed 1 day before manipulation and were starved after amputation. When the tentacle was amputated along with the bulb, the medusae were fed 1 day before manipulation and were fed 3 times a week afterward.

Bright field pictures of medusae were taken with a LEICA S8APO microscope with a Nikon digital camera (D5600) or a Zeiss AXIO Zoom.V16.

## Immunofluorescence

The medusae were anesthetized with 7% $MgCl_2$ in deionized water for 5 min and fixed at room temperature (RT) for 1 h or overnight at 4˚C with 4% paraformaldehyde (PFA) in ASW. After fixation, the samples were washed 3 times (10 min each) with PBS containing 0.1% Triton X-100 (0.1% PBT). The samples were incubated in primary antibodies in 0.1% PBT overnight at 4˚C. The antibodies used were mouse anti-acetylated tubulin (1:500; Sigma-Aldrich, T6793), rabbit anti-FMRFamide (1:1,000; ImmunoStar, 20091), rabbit anti-Phospho-Histone H3 (Ser10) (1:500; Upstate, 06–570), mouse anti-β-catenin (8E4) (1:100; Enzo, ALX-804-260-C100), and rabbit anti-PKC ζ C20 (1:100; Santa Cruz, sc-216). After primary antibody incubation, the samples were washed 3 times (10 min each) with 0.1% PBT. The samples were incubated with secondary antibodies (1:500; ALEXA FLUOR 488, 555, 647 Donkey or Goat anti-mouse IgG, ALEXA FLUOR 488, 555, 647 Donkey or Goat anti-rabbit IgG; Thermo Fisher Scientific) and Hoechst 33342 (1:500; Thermo Fisher Scientific, H1399) in 0.1% PBT for 1 h in the dark. After 3 washes (10 min each) in 0.1% PBT, the samples were mounted on slides with 70% glycerol. Nuclear and poly-γ-glutamate were stained with DAPI (1:250; Invitrogen, D1306), and actin fibers were stained with Alexa 546 phalloidin (1:400; Invitrogen, A22283) in 0.1% PBT for 1 h. Confocal images were collected through a Zeiss LSM 880 confocal microscope. Image processing (color display) and quantification were performed using ImageJ/Fiji software.

## EdU labeling

The medusae were incubated with 150 μm 5-ethynyl-2′-deoxyuridine (EdU) (EdU kit; Invitrogen, C10337) in ASW for 1 h or 24 h. In chase labeling, the medusae were washed with ASW at least twice (up to 3 times, depending on amount). In pulse labeling, after EdU treatment, the medusae were anesthetized with 7% $MgCl_2$ in deionized water and fixed 4% PFA in ASW. After fixation, the samples were washed and incubated with an EdU reaction cocktail (1× reaction buffer, $CuSO_4$, Alexa Fluor azide 488 or 647, and 1× reaction buffer additive; all included in EdU kit; Invitrogen, C10337 or C10340) for 30 min in the dark. After the EdU reaction, the samples were washed and incubated with Hoechst 33342 in 0.1% PBT for 30 min.

In combination with antibody staining, EdU reaction was performed after incubation with the secondary antibody. To combine EdU labeling with fluorescent in situ hybridization (FISH), the EdU reaction was performed after the incubation with Cy5-tyramide solution.

## EdU and BrdU dual labeling

The medusae were incubated with 150 μm EdU in ASW for 1 h or 24 h before amputation, depending on the experiments. Just after incorporation of EdU, the medusae were washed in

ASW 3 times, and tentacles were amputated. At 24 hpa, the medusae were incubated with 2.5 mM BrdU (abcam, ab142567) in ASW for 1 h. The medusae were washed out in ASW 3 times.

The medusae were anesthetized with 7% MgCl$_2$ for 5 min and fixed with 4% PFA in ASW. The samples were washed 3 times with 0.1% PBT and incubated with an EdU reaction cocktail for 30 min in dark. The samples were washed 3 times and treated with 2N HCl for 30 min and then washed 3 times and incubated in primary antibodies in 0.1% PBT overnight at 4˚C. The primary antibody used was mouse anti-BrdU (1:100; BD, 347580). The samples were washed 3 times and incubated with secondary antibodies (1:500; ALEXA FLUOR 555 Goat anti-mouse IgG) and Hoechst 33342 in 0.1% PBT for 1 h in the dark and washed 3 times.

## Fluorescent in situ hybridization

The following FISH protocol was created from the published protocol [38]. The sequences of *Nanos1*, *Nanos2*, and *Piwi* have been previously described [39], the presumptive sequence of *Minicollagen1* (*Mcol1*) and *Vasa1* were acquired from the annotation of an RNA-seq result [33]. Purified total RNA was reverse transcribed into cDNA by PrimeScript II 1st strand cDNA Synthesis Kit (Takara, 6210A) or SMARTer RACE cDNA Amplification Kit (Clontech). A cDNA library was used as a template for PCR (*Nanos1*, *Nanos2*, *Piwi*, and *Mcol1*) and 3′ RACE (*Vasa1*). The primer sets used for PCR cloning are as follows: *Nanos1*: 5′-AAGAGACA CAGTCATTATCAAGCGA-3′ (forward) and 5′-AGCACGTAAAATTGGACACGTCG-3′ (reverse), *Nanos2*: 5′- ACTTCTCCAAAACCTCATGCCGAG-3′ (forward) and 5′- GAATGG CGGGCGATTTGACATCC-3′ (reverse), *Piwi*: 5′- CACACAAGAGTTGGACCGGA-3′ (forward) and 5′- ACCGGCTTATCGATGCAACA-3′ (reverse), *Vasa1*: 5′-GCCACCCAAAGAA GACAGACAGACAC-3′ (forward for 3′RACE) and 5′-CGAAACGACTTGCTGATTTTCTC GCCAG -3′ (nested forward for 3′RACE*)*, *Mcol1*: 5′-CTCGTCGGTATTGCCCTCTC-3′ (forward) and 5′-CCAACCTATCGTGGACGTGT-3′ (reverse). PCR products were subcloned into the TAK101 vector (TOYOBO) and RACE product was subcloned into the pGEM-T Easy vector (Promega). The resulting plasmids were used for RNA probe synthesis with digoxigenin (DIG) labeling mix (Roche, 11277073910) or fluorescein (FITC) labeling mix (Roche, 11685619910), and T7 (Roche, 10881767001) or T3 RNA polymerase (Roche, 11031163001) was used, according to the insert direction.

Medusae were anesthetized 7% MgCl$_2$ for 5 min and fixed overnight at 4˚C with 4% PFA in ASW. Briefly, fixed samples were washed 3 times with PBS containing 0.1% Tween-20 (PBST), followed by pre-hybridization in hybridization buffer (HB buffer: 5 × SSC, 50% formamide, 0.1% Tween-20, 50 μg/ml tRNA, 50 μg/ml heparin) at 55˚C for 2 h. The samples were hybridized with HB buffer containing the antisense probes (final probe concentration: 0.5–1 ng/μl in HB buffer) at 55˚C overnight. The samples were washed twice each with wash buffer 1 (5 × SSC, 50% formamide, 0.1% Tween-20), wash buffer 2 (2 × SSC, 50% formamide, 0.1% Tween-20), and 2 × SSC. The samples were then washed with 0.1% PBST and incubated in 1% blocking buffer (1% blocking reagent [Roche] in Maleic acid) for 1 h. The samples were incubated with anti-DIG-POD antibodies (1:500; Roche, 11207733910) in 1% blocking buffer overnight at 4˚C. The samples were then washed with Tris-NaCl-Tween Buffer and incubated with Cy5-tyramide solution (TSA Plus Cyanine 5; AKOYA Biosciences, NEL745001KT) for 10 min. Finally, the samples were washed with 0.1% PBST and incubated with Hoechst 33342 in 0.1% PBST for 30 min in the dark and washed 3 times.

For double FISH, DIG for low expression genes and FITC probes for high expression genes were used (S3E Fig; *Nanos1*-DIG and *Nanos2*-FITC, S3F Fig; *Mcol1*-DIG and *Nanos2*-FITC). The samples were then washed with Tris-NaCl-Tween Buffer after incubation with anti-DIG--POD antibodies and incubated with Cy3-tyramide solution (TSA Plus Cyanine 3; AKOYA

Biosciences, NEL744001KT) for 10 min. The samples were washed with 0.1% PBST and incubated with 3% $H_2O_2$ for 15 min. The samples were washed with 0.1% PBST and incubated with anti-Fluorescein-POD antibodies (1:500; Roche, 11426346910) in 1% blocking buffer overnight at 4°C. The samples were then washed with Tris-NaCl-Tween Buffer and incubated with Cy5-tyramide solution for 10 min. Finally, the samples were washed with 0.1% PBST and incubated with Hoechst 33342 in 0.1% PBST for 30 min in the dark and washed 3 times.

### DiI injection

Medusae were relaxed in 7% $MgCl_2$ on a petri dish, and 10 mM CellTracker CM-DiI (Invitrogen, C7001) was injected into the epidermal layer of the tentacle bulb or the center of the tentacle, using a micro injector (Eppendorf, Femtojet 4i). The quartz capillary (Sutter Instrument, QF100-70-10) was pulled by Laser-Based Micropipette Puller P2000 (Sutter Instrument). After injection, injected tentacles were amputated. To identify DiI-labeled cells, medusae were incubated with 150 μm EdU for 1 h before DiI injection or antibody staining was performed after DiI injection using anti-β-catenin. To monitor DiI-labeled cells during blastema formation, at 24 hpa, the injected medusae were incubated with 150 μm EdU in ASW for 1 h. The medusae were relaxed in 7% $MgCl_2$ and fixed in 4% PFA. After washing in 0.1% PBT, the medusae were incubated with an EdU reaction cocktail for 30 min in the dark. After the EdU reaction, the samples were washed and incubated with Hoechst 33342 in 0.1% PBT.

### X-ray irradiation

The medusae were placed in a V7 plastic container with 6 to 8 ml of ASW, the minimum required for their maintenance. X-ray irradiations were performed using an X-ray machine (mediXtec, MX-160Labo). Medusae were positioned 19 cm from the X-ray source, and a dose of 30 Gy (19.3 min at 160 kV 3 mA), 50 Gy (32.2 min at 160 kV 3 mA), or 75 Gy (48.3 min at 160 kV 3 mA) was delivered. After irradiation, the medusae were moved to a plastic container with fresh ASW.

### Drug treatment

The medusae were incubated with 10 mM hydroxyurea (HU) (085–06653; Wako, Osaka, Japan) or 30 μm Mitomycin C (139–18711; Wako, Osaka, Japan) in ASW (ASW only for control). Drug incubation was sustained over 3 days. Medusae were fed every other day, and the drug solutions were renewed after feeding.

### qPCR

Total RNA was purified from the whole tentacles of 3 medusae with RNeasy Mini kits (Qiagen). Lysate was treated with Dnase I (Qiagen, 79254) for 15 min at RT. cDNA was synthesized from 200 ng of total RNA by Prime script RT master Mix (Takara, RR036A). RT-qPCR was performed using TB Green Premix Ex TaqII (Tli RnaseH Plus) (Takara, RR820L) and a Quantstudio6 Flex Real-Time PCR system (Thermo Fisher) using *F-actin capping protein subunit beta* (*CpCapZbeta*) as an internal control [32]. The qPCR primer sets are as follows: *CpCapZbeta*: 5′-AAAGAAAGCTGGAGACGGTTCA-3′ (Forward), 5′-GTAGTGGGCATTT CTTCCGC-3′ (Reverse), *Nanos1*: 5′-TTCGTCAAGTGGCAGTCGTG-3′ (Forward), 5-CAAGCCCTGGTACAAACGGA-3′ (Reverse).

### UV laser exposure

Tentacles were amputated at 1 day before UV exposure. At 24 hpa, the medusae were incubated with Hoechst 33342 (1:250) for 20 min and washed 3 times in ASW. After washing, the

relaxed medusae were placed on slides with 7% $MgCl_2$ and enclosed with cover glass. UV laser was exposed to the blastema region of the amputated tentacle using the photo bleaching application (50 cycles at 100% 405 laser power) of the Zeiss LSM 880 confocal microscope. The bleached region is shown with a white dot square (400 × 400 pixels, W × H) in S11B Fig. After UV exposure, the medusae were moved to ASW in plastic containers and monitored to examine the effect on regeneration.

### Quantification and statistical analysis

**Distribution of cells.** Three areas were defined (the basal, middle, and distal regions) by dividing the tentacle into 3 equal lengths in confocal images (Figs 2A, 2B, 3C, 3D, S2A and S2B). The number of PH3$^+$, EdU$^+$, or *Nanos1$^+$* cells was counted manually in each area using the multipoint tool of ImageJ/Fiji.

**Counting the number of cells.** The PH3$^+$, EdU$^+$, or *Nanos1$^+$* cell ratio (Figs 2D–2F, 4I, 4J, S7B, S7C, S7E and S7F) was measured by counting the total number of cells as well as the PH3$^+$, EdU$^+$, or *Nanos1$^+$* cells using ImageJ/Fiji. Each signal$^+$ cell was counted manually using the multipoint tool of ImageJ/Fiji. Total cell number quantification was performed as follows: (1) Binarize the signal of Hoechst staining using the "Threshold" command. (2) Divide continuously adjacent multiple nuclei using the "Watershed" command. (3) Measure the number of nuclei using the "Analyze Particles" command.

Other cell number counting (Figs 3A, 3B, 3E, 3F, 4C–4F, 5B, 5C, 5F, 5G, 6B, 6C, 6E, 6F, S2C, S2E, S4C–S4E, S5A–S5D, S6B, S6C, S8A, S8B, S10, S11C and S11D) was performed manually using the multipoint tool of ImageJ/Fiji.

**Measurement of tentacle length and area, signal intensity.** The tentacle length (Fig 5K and 6I) was measured using the segment line tool of ImageJ/Fiji. The tentacle area (S5C–S5E Fig) was recorded as the ROI with the polygon selection tool and measured by ImageJ/Fiji. The quantification area was marked with rectangle tool, and the signal intensity was measured by ImageJ/Fiji (S2D Fig).

**Statistics.** Statistical analyses were performed using Excel and Graphpad Prism9. Two tailed *t* tests were used for comparisons between 2 groups. Significance is indicated in the figures as follows: $^*P \leq 0.05$, $^{**}P \leq 0.05$, $^{***}P \leq 0.001$, Not Significant (NS): $P > 0.05$. Bar graphs show mean ± standard error. Dots in bar graphs and boxplots indicate individual values.

## Results

For cnidarians, tentacles are essential and common appendages for capturing prey and defending against predators. Cnidarian polyps such as *Hydra*, *Hydractinia*, and *Nematostella* have tentacles in the head region, near the hypostome, which can restore functional tentacles during head regeneration [18,20,40]. Similarly, studies using hydrozoan jellyfish have shown that medusae tentacles can regenerate upon amputation [24,27]. Although cnidarians commonly exhibit high regenerative capacity for tentacles, the exact cellular processes and the limitation of regeneration are poorly understood.

To monitor the process of tentacle regeneration and to explore its responses to different dissections, we utilized the medusa tentacle of the hydrozoan jellyfish *Cladonema* (Fig 1). In *Cladonema* tentacles, proliferative cells including stem cells, or i-cells, are localized on the basal side, called the tentacle bulb, while nematocytes are distributed as clusters on the distal side, similar to the jellyfish *Clytia* (Figs 1C, 2Ai, 3Ci, S1A, S3A–S3D, S4Ai, and S4Bi) [24,37,39]. The existence of localized stem cells at the bulb prompted us to examine tentacle regeneration as a model for appendage regeneration in non-bilaterians.

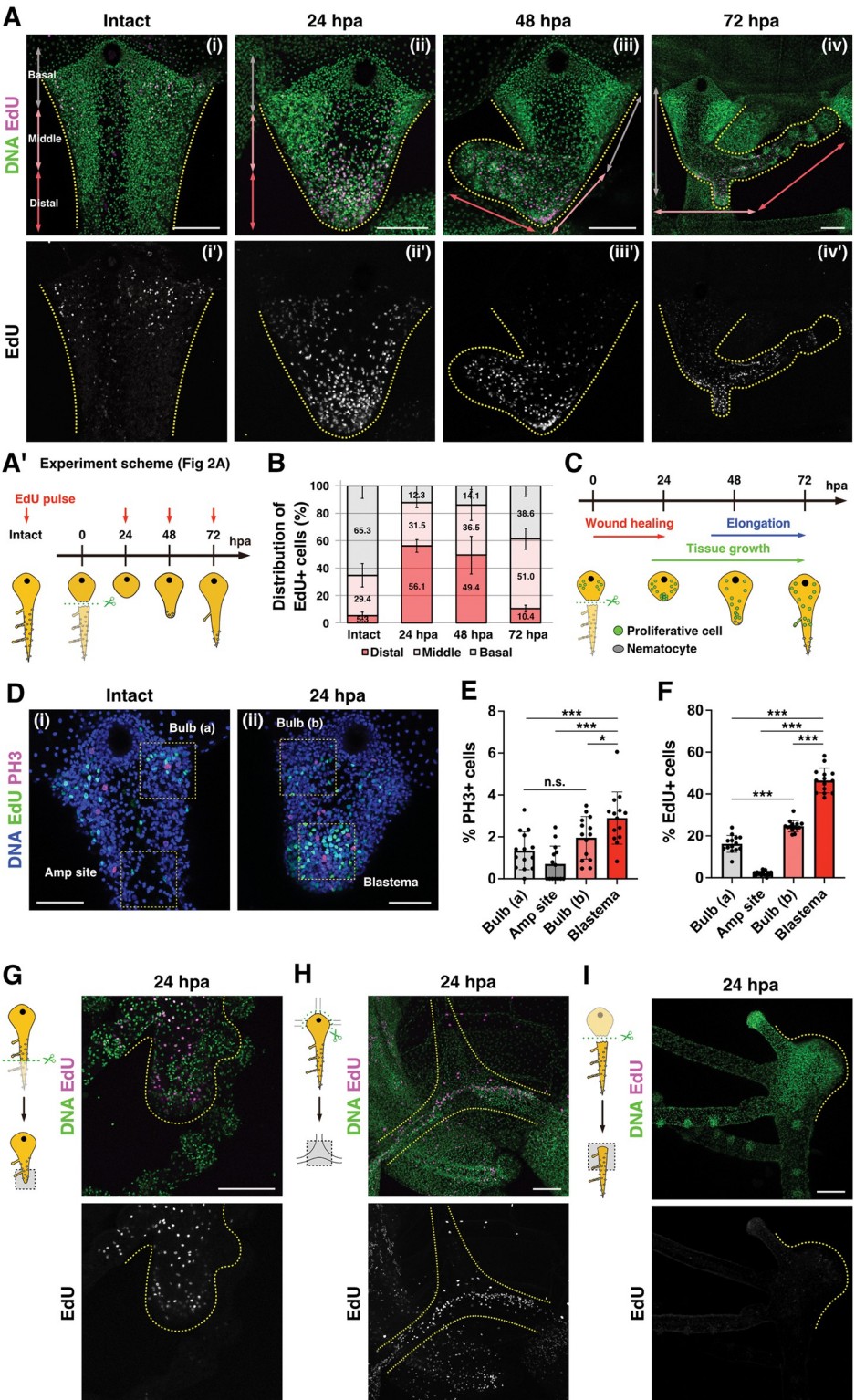

**Fig 2. Blastema formation at early phase of tentacle regeneration.** (A) Distribution of S-phase cells in intact and regenerating tentacle stained with EdU 1 h pulse labeling at intact as well as 24, 48, and 72 hpa, as shown in the scheme (A'). The tentacles were divided into 3 parts defined as basal (gray arrows), middle (pink arrows), and distal (red arrows). (B) Quantification of EdU⁺ cells' distribution in regenerating tentacle based on the defined areas in (A). Intact: *n* = 19 (tentacles), 24 hpa: *n* = 7, 48 hpa: *n* = 10, 72 hpa: *n* = 6. (C) Scheme of proliferative cells distribution

during tentacle regeneration. (D) Distribution of S-phase and M-phase cells stained with EdU 1 h pulse labeling and anti-Phospho-Histone 3 (PH3) antibody. EdU (green) and PH3 (magenta). Yellow dot squares show each quantification area ($74.72^2$ $\mu m^2$) in (E, F). (E, F) The proportion of PH3$^+$ and EdU$^+$ cells per all cells in each area. The regions of bulb (a) and amputation (amp) site were defined in intact tentacles while those of bulb (b) and blastema were defined in the regenerating tentacle at 24 hpa. Each area: $n = 14$. (G–I) Distribution of proliferative cells at 24 hpa: (G) After the distal side of tentacle is amputated, (H) after removing the bulb from canals, and (I) in isolated tentacles at 24 hpa. EdU 1 h pulse labeling at 24 hpa. The numerical values that were used to generate the graphs in (**B, E, F**) can be found in S1 Data. Unpaired two-tailed $t$ test. $^*p < 0.05$, $^{***}p < 0.001$. Scale bars: (A, G–I) 100 $\mu m$, (D) 50 $\mu m$. hpa, hours post-amputation.

## *Cladonema* medusa tentacles functionally regenerate upon amputation

When the tentacle was amputated with the bulb left intact (Fig 1D and 1E), wound closure finished by 24 hours post-amputation (hpa) (S1B and S1C Fig), at which point the injury site became smooth surface (Fig 1Di and 1Dii). Subsequently, tentacles began to elongate distally, and nematocyte clusters, the spherical structures stained with the mature nematocyte marker poly-γ-glutamate, formed at the tip of the regenerating tentacle at 48 hpa (Figs 1Diii, 1Fiii, and S1A). The number of nematocyte clusters increased during tentacle elongation, and these clusters were localized along the distal side at 72 hpa, which is when the regenerating tentacle branched, a characteristic feature of the genus *Cladonema* (Fig 1Div and 1E). Notably, these regeneration processes occurred without feeding, suggesting that tentacle regeneration is an immediate response for the organism to re-construct the organ that is required for capturing food.

To investigate the subtleties of tentacle morphology at cellular resolution during regeneration, we visualized the muscle fibers, mature nematocytes, and neurons by staining F-actin, poly-γ-glutamate, and FMRFamide and acetylated tubulin, respectively (Figs 1F, 1G, S1D, and S1E). During the early stages of regeneration (0 to 24 hpa), wound closure occurred progressively via the accumulation of supra-cellular actin fibers (Fig 1Fi and 1Fii). Accordingly, muscle fibers regenerated during tentacle elongation (Fig 1Fiii and 1Fiv). While mature nematocytes were rarely found in the bulb or the injury site until 24 hpa (Fig 1Fi and 1Fii), nematocytes began to accumulate at the tip of the regenerating tentacle at 48 hpa and formed clusters at 72 hpa (Fig 1Fiii and 1Fiv). The re-formation of neurons on the distal side occurred at 48 hpa and completed at 72 hpa (Figs 1G, S1D, and S1E). These results suggest that the regenerating tentacle is fully functional after 48 hpa. To test this possibility, we examined whether the regenerating tentacle is capable of feeding. While the regenerating tentacle was not functional until 24 hpa, more than half of the regenerating tentacles captured prey at 48 hpa (55%), and this ratio reached 100% at 72 hpa (Fig 1H and 1I). These results indicate that *Cladonema* medusae can fully regenerate a functional tentacle within 2 to 3 days of amputation.

While the *Cladonema* medusa tentacle can almost always regenerate with the bulb left intact, it is unclear whether the tentacle can still regenerate even after the bulb has been removed (Fig 1Ji). We thus ablated the entire tentacle, including the bulb, and monitored the regeneration process. After amputation, the ring canal and the radial canal adhered, and the wound closure completed at 1 day post-amputation (dpa) (Fig 1Jii). The injured area began to bulge as early as 5 dpa, and the transparent wound site turned orange, which reflects the color of its food, *Artemia* (Fig 1Jiii). The de novo tentacle bulb reformed within 1 to 2 weeks with a nematocyte cluster at the tip, and the small tentacle gradually elongated (Fig 1Jiv and 1K). Because the rate of successful regeneration varied dramatically while rearing medusae under the same conditions and feeding frequency, tentacle bulb regeneration likely depends on the health condition of the individual medusa (S1F Fig).

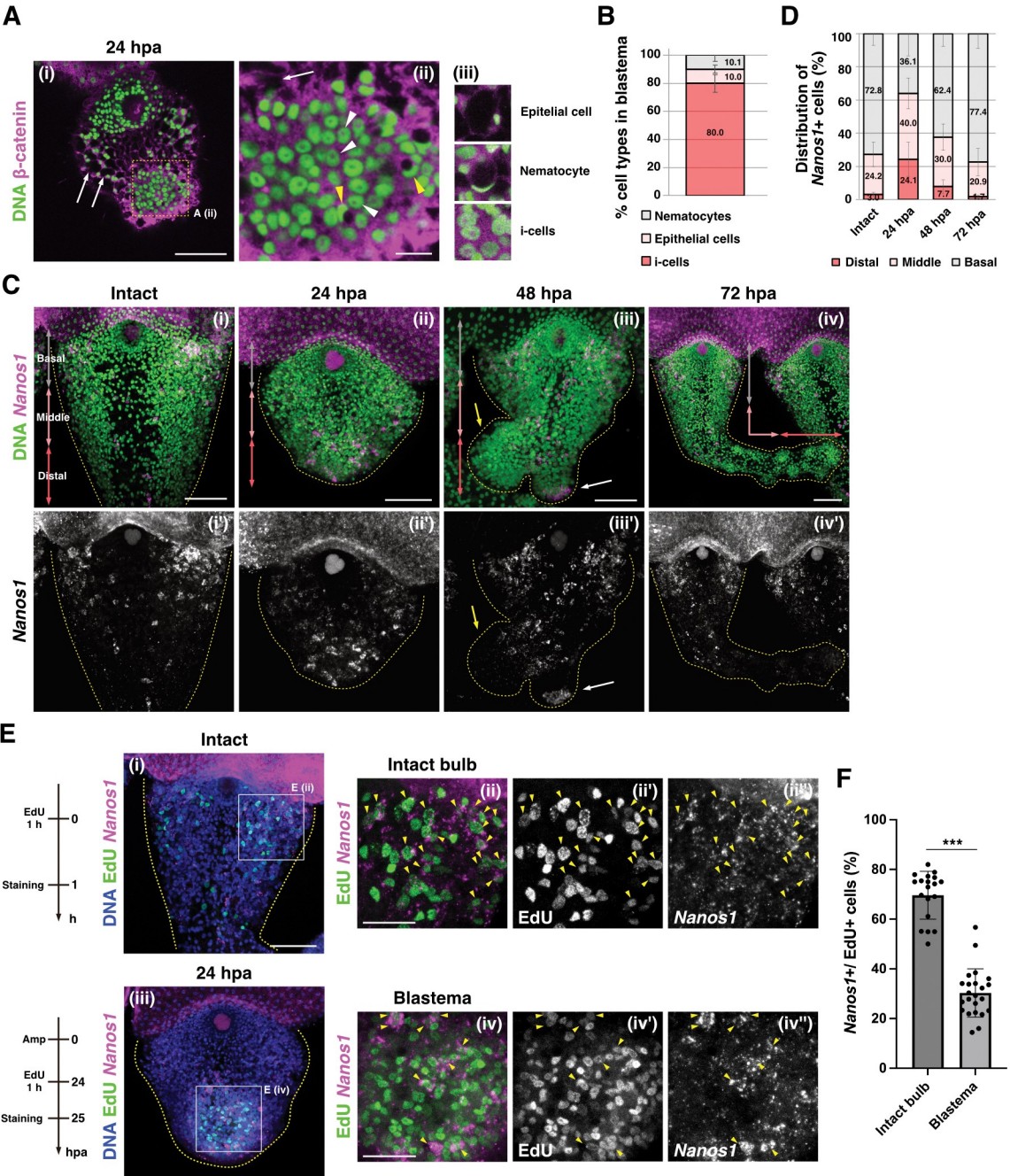

**Fig 3. Stem cells and progenitor cells in blastema.** (A) Identification of cell types in blastema stained with anti-β-catenin staining. White arrow: epithelial cell, white arrow head: i-cell, yellow arrow head: nematocyte. Yellow dot square (ii) shows quantification area ($35.10^2$ μm$^2$) in (B). Magnified images for each cell type (iii). (B) Cell types in blastema at 24 hpa by anti-β-catenin staining. $n = 3$ (areas). (C) FISH images of *Nanos1* gene expression from the proximal, abaxial side of the tentacle. Distribution of stem (*Nanos1+*) cells in intact and regenerating tentacle. The tentacles were divided into 3 parts defined as basal (gray arrows), middle (pink arrows), and distal (red arrows). Yellow arrow indicates main tentacle and white arrow indicates new branch. (D) Quantification of *Nanos1+* cells distribution in tentacles based on the defined areas in (C). Intact: $n = 5$ (tentacles), 24 hpa: $n = 4$, 48 hpa: $n = 3$, 72 hpa: $n = 5$. (E) Images of the intact and regenerating tentacle co-labeled with *Nanos1* FISH and EdU 1 h pulse labeling. EdU (green) and *Nanos1* (magenta). White squares show the quantification area ($74.72^2$ μm$^2$) in (3F and S4E). Yellow arrowheads indicate EdU$^+$ and *Nanos1+* cells. (F) The ratio of *Nanos1+* cells/EdU$^+$ cells in intact and regenerating tentacles. The averages of *Nanos1+* cells/EdU$^+$ cells, intact bulb: 69.7% ($n = 18$), blastema: 30.3% ($n = 22$). The numerical values that were used to generate the graphs in (**B, D, F**) can be found in S1 Data. Unpaired two-tailed *t* test. ***$p < 0.001$. Scale bars: (Ai, C, Ei, and Eiii) 50 μm, (Eii and Eiv) 25 μm, (Aii) 10 μm. FISH, fluorescent in situ hybridization; hpa, hours post-amputation.

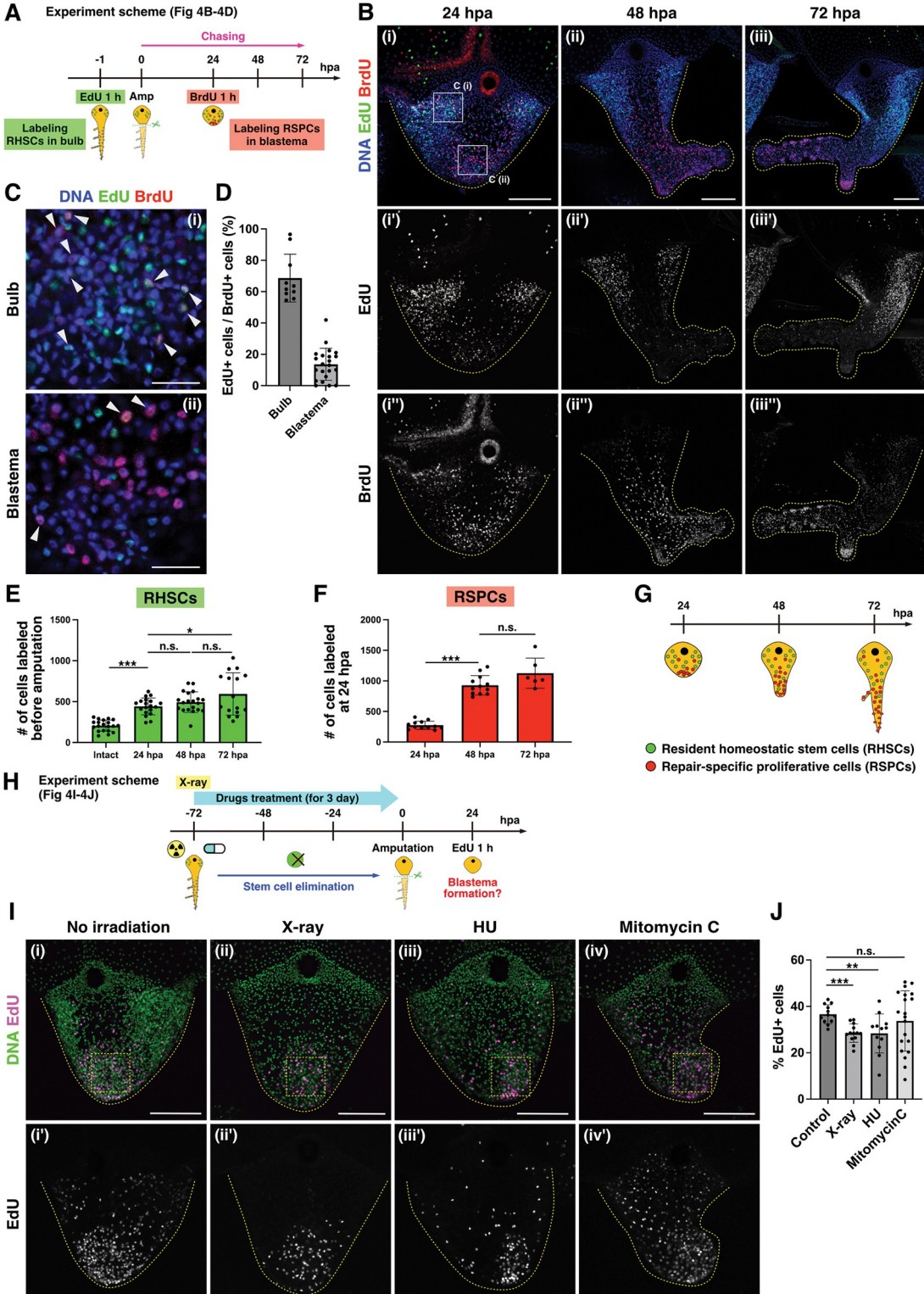

**Fig 4. Distinct proliferative cell populations during tentacle regeneration.** (A) Scheme of EdU and BrdU chase experiments in (B–D). RHSCs are labeled by EdU before amputation; RSPCs are labeled by BrdU at 24 hpa. (B) Distribution of EdU$^+$ and BrdU$^+$ cells during tentacle regeneration. EdU (green) and BrdU (red). (C) The magnified images of bulb and blastema in (Bi). White arrowheads indicate EdU$^+$ and BrdU$^+$ cells. (D) Quantification of EdU$^+$/BrdU$^+$ cells in (C). Each quantification area size is 74.72$^2$ μm$^2$. The ratio of EdU$^+$/BrdU$^+$ cells. bulb: $n$ = 10, blastema: $n$ = 22. (E) The number of RHSCs during regeneration. Intact: $n$ = 20

(tentacles), 24 hpa: $n = 19$, 48 hpa: $n = 20$, 72 hpa: $n = 16$. (F) The number of RSPCs during regeneration. 24 hpa: $n = 12$, 48 hpa: $n = 12$, 72 hpa: $n = 6$. For (E) and (F), quantification area is the whole regenerating tentacle. (G) Scheme of 2 proliferative cell populations during regeneration. (H) Scheme of stem cell elimination by X-ray irradiation (75 Gy) or drug treatments (HU 10 mM and Mitomycin C 30 μm) followed by labeling blastema with EdU (I, J). Tentacle amputation at 3 days post-irradiation/after treatment of drugs. (I) Distribution of proliferative cells after X-ray irradiation or drug treatments. EdU 1 h pulse labeling at 24 hpa. Yellow dot squares show each quantification area ($74.72^2$ μm$^2$) in (J). (J) The ratio of EdU$^+$ cells in blastema. Control: $n = 12$, X-ray: $n = 13$, HU: $n = 12$, Mitomycin C: $n = 20$. The numerical values that were used to generate the graphs in (**D–F, J**) can be found in S1 Data. Unpaired two-tailed $t$ test. $^*p < 0.05$, $^{**}p < 0.005$, $^{***}p < 0.001$. Scale bars: (B, I) 100 μm, (C) 20 μm. hpa, hours post-amputation; RHSC, resident homeostatic stem cell; RSPC, repair-specific proliferative cell.

We also examined the regenerative potential of isolated tentacles. While the whole body cannot be regrown from isolated tentacles in *Hydra* or *Nematostella* [21,41], polyps in the jellyfish *Aurelia aurita* can regenerate from isolated tentacles [42]. To examine the regeneration capacity of the isolated tentacle in *Cladonema*, we cultured *Cladonema* medusa tentacles either with or without the bulb left intact. Although wound closure successfully occurred regardless of the existence of the bulb, tissues, and organs located to the proximal side of the tentacle including the umbrella, the radial canal, and the bulb, did not regenerate, and the isolated tentacle gradually degenerated (S1G and S1H Fig).

These observations together show that the *Cladonema* medusa tentacle can functionally regenerate after amputation, but that its organ repair capacity is restricted to the organism that recognizes the missing tentacle, likely through the canals, as previously shown in the *Clytia* medusa [27]. For our remaining studies, we focused our attention on the process of tentacle regeneration with the bulb intact, which provides a highly reproducible and tractable system.

## Blastema is formed in the early phase of tentacle regeneration

During organ regeneration, blastema forms at the regenerating tip after wound closure and contributes to reconstruction of the new tissue as a source of cell proliferation and differentiation [1,2]. In order to investigate the timing of blastema formation during *Cladonema* tentacle regeneration, we performed pulse labeling of EdU, an S-phase marker (Fig 2A) [43]. To understand the spatial pattern of cell proliferation, we defined 3 areas in the regenerating tentacle (the basal, middle, and distal regions) and quantified the distribution of proliferating cells. Consistent with the localization of stem cells (Figs 3Ci, S4Ai, and S4Bi) [39], cell proliferation in the intact tentacle was mainly detected on the basal side (Fig 2Ai and 2B). By contrast, cell proliferation in the regenerating tentacle was frequently observed on the distal side near the wound area at 24 hpa (Fig 2Aii and 2B). At 72 hpa, unlike at 24 and 48 hpa, cell proliferation was detected in the middle and on basal side, not on the distal side (Fig 2Aiv and 2B). To investigate the exact timing of the cell proliferation increase at the injury site, we examined the localization of EdU$^+$ cells every 4 h immediately after amputation to 24 hpa (S2C Fig). We found that the number of proliferating cells significantly increased at 20 hpa and peaked at 24 hpa (S2D and S2E Fig). We further confirmed the pattern of proliferating cells using the M-phase marker, anti-Phospho-Histone 3 (PH3) antibody. While PH3$^+$ cells were distributed across the basal side before amputation, in the regenerating tentacle, PH3$^+$ cells accumulated on the distal side at 24 hpa, and gradually localized on the basal side starting at 48 hpa (Figs 2D, S2A, and S2B), which is consistent with the results of EdU$^+$ cell distribution (Fig 2A–2C). These results also reflect the fact that differentiated nematocytes accumulate on the distal side at 48 hpa and 72 hpa (Fig 1D and 1F). Together, these observations suggest that cell proliferation is accelerated near the injury site at 24 hpa.

Blastema cells are undifferentiated cell populations that possess a high mitotic capacity during regeneration [44]. We thus investigated proliferative activity in detail during the early

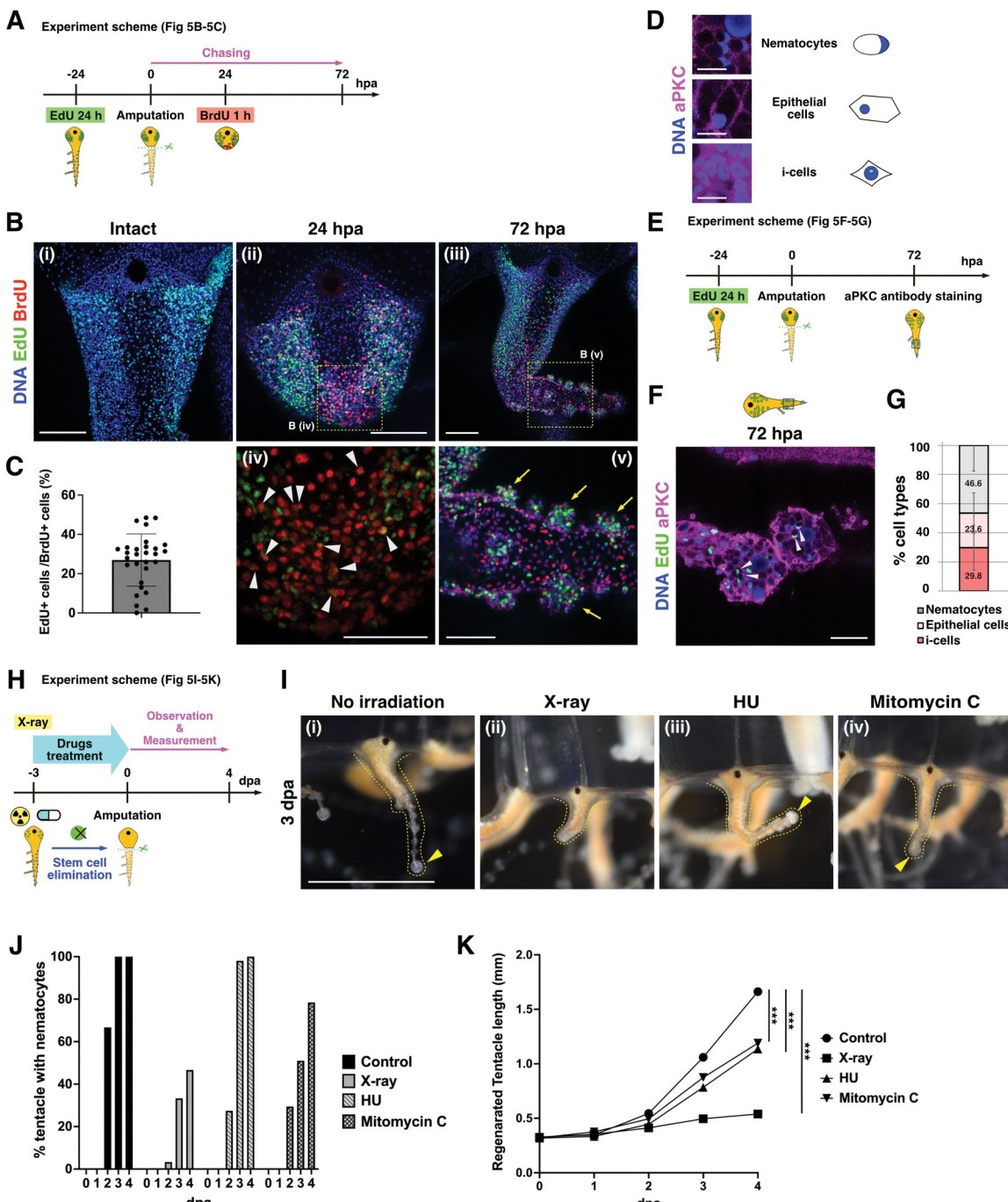

**Fig 5. Contribution of RHSCs to nematogenesis and tentacle elongation.** (A) Scheme of EdU long-time (24 h) and BrdU double chasing experiment in (B, C). (B) The distributions of EdU+ and BrdU+ cells during tentacle regeneration. EdU (green) and BrdU (red). Yellow arrows indicate accumulation of EdU+ cells at nematocyte clusters in distal side at 72 hpa. White arrowheads indicate EdU+/ BrdU+ cells. Yellow dot squares show magnified views (iv, v). (C) Quantification of EdU+/BrdU+ cells (%) in the blastema region (Biv). The quantification area is $74.72^2$ μm². $n = 29$. (D) Cell type identification by anti-aPKC staining. i-cells, nematocytes, and epithelial cells, similar to the anti-β-catenin staining (see Fig 3A). (E) Scheme of experiments (F, G). RHSCs are labeled by EdU 24 h incorporation before amputation and anti-aPKC staining at 72 hpa. (F) Cell type identification of EdU-labeled cells in distal side at 72 hpa by anti-aPKC staining. EdU (green) and aPKC (magenta). White arrowheads show nematocytes. (G) Quantification of the cell types of EdU+ cells on the distal side of a regenerating tentacle at 72 hpa. Quantification area of regenerating tentacle is $74.72^2$ μm². $n = 10$. (H) Experimental scheme for tentacle regeneration after X-ray irradiation (75 Gy) or drug treatments (HU 10 mM and Mitomycin C 30 μm) in (I–K). Tentacle amputation at 3 dpi or after 3 days treatment of drugs. (I) Representative images of regenerating tentacles at 3 dpa. Yellow arrowheads indicate nematocyte clusters. (J) The timing of nematocyte cluster formation in regenerating tentacles. Control:

$n$ = 48, X-ray: $n$ = 60, HU: $n$ = 51, Mitomycin C: $n$ = 51. (K) Length of regenerating tentacle after irradiation and drug treatments. Control: $n$ = 48, X-ray: $n$ = 60, HU: $n$ = 51, Mitomycin C: $n$ = 51. The numerical values that were used to generate the graphs in (**C, G, J, K**) can be found in S1 Data. Unpaired two-tailed $t$ test. ***$p$ < 0.001. Scale bars: (I) 1 mm, (Bi–iii) 100 μm, (Biv and Bv) 50 μm, (F) 25 μm. dpa, day post-amputation; dpi, days post-irradiation; hpa, hours post-amputation; RHSC, resident homeostatic stem cell.

phase of tentacle regeneration (Fig 2D–2F). In the regenerating tip, the number of PH3$^+$ and EdU$^+$ cells was about 4 and 20 times greater, respectively, than the corresponding area (amputation site) in the intact tentacle (Fig 2D–2F). Moreover, these PH3$^+$ and EdU$^+$ cell numbers at the regenerating tip were significantly greater than those in the bulb of the intact tentacle where resident stem cells are localized (Fig 2D–2F). Notably, nearly half of the cells in the regenerating tip are EdU$^+$ cells (Fig 2Dii and 2F). Altogether, these results indicate that proliferative activity in the regenerating tip is much higher than that of cells not in the regenerating tip, which provides evidence that blastema cells appear during tentacle regeneration.

To determine whether blastema formation is specific to the location of amputation, we examined the localization of proliferating cells after dissecting tentacles in different ways. When the distal side of the tentacle is amputated, ectopic cell proliferation was observed near the injury site at 24 hpa (Fig 2G). Similarly, when the tentacle was amputated with the bulb, cell proliferation occurred around the injury site where the canals merged at 24 hpa (Fig 2H). These results suggest that blastema is formed around the injury site regardless of the location of amputation.

Based on these observations, we hypothesized that blastema is a prerequisite for proper regeneration of the tentacle. To test this possibility, we examined whether blastema formation occurs in the isolated tentacle, which fails to regenerate basal tissue (S1G and S1H Fig). In the isolated tentacle, EdU$^+$ cells were not detected at the injury site at 24 hpa (Fig 2I). These results together suggest that blastema formation is associated solely with the regenerating tentacle.

## Stem cells compose blastema

What cell types constitute blastema? In hydrozoans, i-cells behave as pluri-/multipotent stem cells that can differentiate into progenitors and differentiated lineages [45]. Indeed, i-cells in hydrozoan jellyfish including *Clytia* and *Cladonema*, which are mainly localized at the tentacle bulb, are defined by (1) morphological features like a large nucleus-to-cytoplasm ratio with prominent nucleoli; (2) the expression of stem cell marker genes such as *Nanos1* and *Piwi*; and (3) the ability to self-renew and differentiate into multiple cell types [39,46,47]. It is thus possible that undifferentiated stem cells like i-cells contribute to blastema. In order to confirm the cell types that form blastema, we utilized the combination of β-catenin antibody staining and nuclear staining, which can distinguish i-cells (including i-cells and early nematoblasts), nematocytes, and epithelial cells (Fig 3A). In both *Hydractinia* and *Cladonema*, i-cells were identified by cytoplasmic signals of β-catenin, together with the large nucleus-to-cytoplasm ratio and prominent nucleoli [39,48]. While nematocytes exhibit crescent-shaped nuclei, epithelial cells show a polygonal shape delineated by the membrane-localized β-catenin (Fig 3Aiii). Through the quantification of cell types, we found that around 80% of blastema cells were i-cells, 10% were epithelial cells, and 10% were nematocytes (Fig 3A and 3B), indicating that most cells in blastema are morphologically i-cells.

To molecularly characterize these blastema cells, we examined the expression of the stem cell markers, *Nanos1*, *Piwi*, and *Vasa1* by FISH. Note that, while *Cladonema* possess 2 *Nanos* genes, *Nanos1* and *Nanos2*, *Nanos2*$^+$ cells co-express a nematoblast marker *Mcol1*, suggesting that *Nanos2* is a nematoblast marker rather than a stem cell marker in *Cladonema* medusa (S3E and S3F Fig), similar to *Hydractinia* [49]. In the intact tentacle, *Nanos1*$^+$, *Piwi*$^+$, or

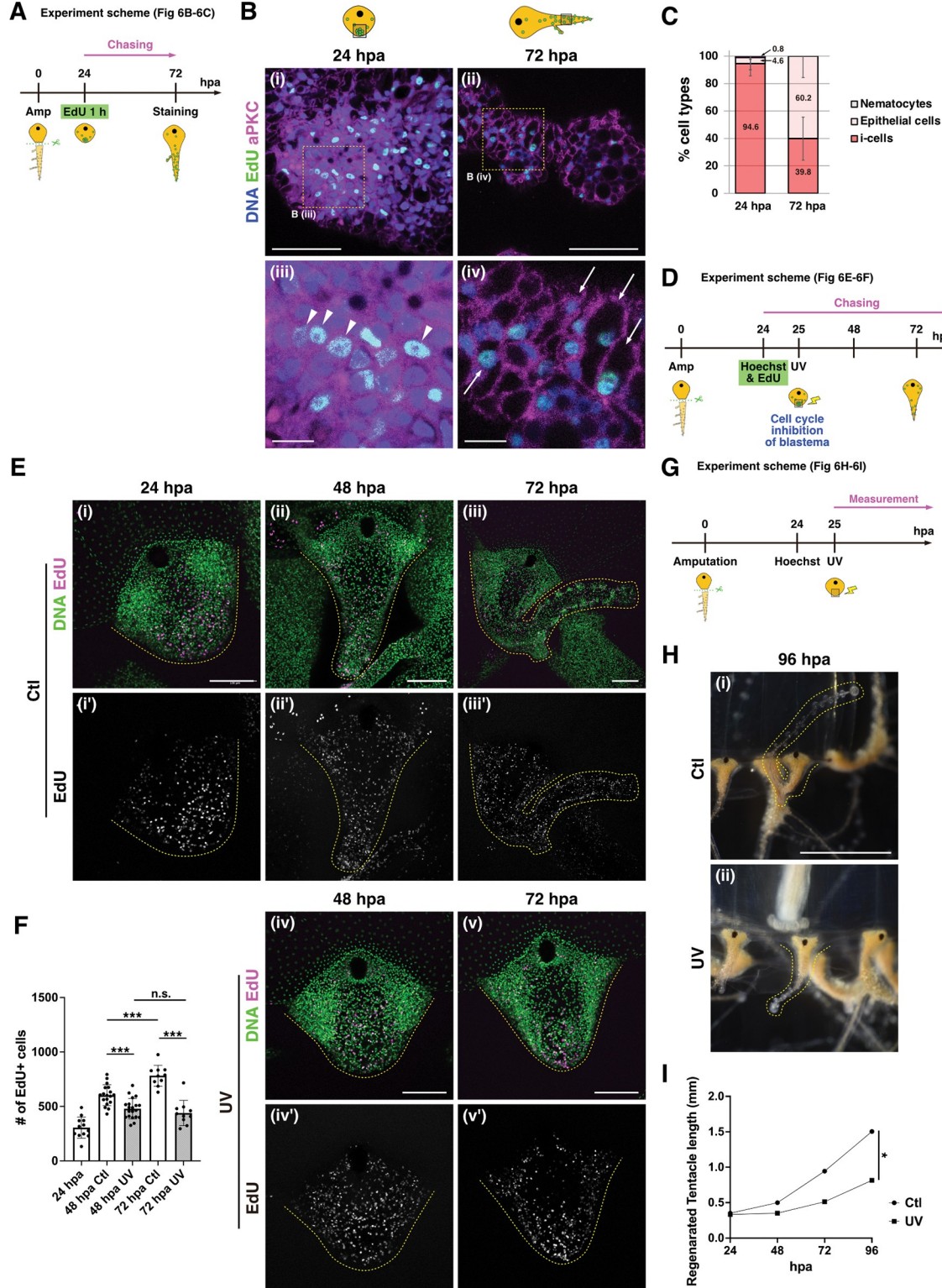

**Fig 6. Contribution of RSPCs to tentacle elongation.** (A) Experimental scheme for combining the identification of cell types by anti-aPKC staining and EdU chasing in (B and C). (B) Identification of cell types of EdU+ cells in blastema at 24 hpa and the distal side of the regenerating tentacle at 72 hpa by anti-aPKC staining. EdU (green) and aPKC (magenta). White arrow: epithelial cell, white arrow head: i-cell. Magnified views (iii, iv). (C) Quantification of cell types in EdU+ cells at 24 and 72 hpa. Quantification area of regenerating tentacle is $35.06^2$ μm$^2$; 24 hpa: $n = 9$, 72 hpa: $n = 12$. (D) Experimental scheme depicting blastema monitoring after

UV exposure in (E and F). EdU 1 h incorporation at 24 hpa and UV exposure around the blastema region. (E) Distribution of EdU-labeled cells during regeneration, Ctl (no UV) vs. UV. (F) The number of EdU$^+$ cells from 24 hpa to 72 hpa, Ctl (no UV) vs. UV; 24 hpa: $n = 12$ (tentacles), 48 hpa Ctl: $n = 18$, 48 hpa UV: $n = 20$, 72 hpa Ctl: $n = 10$, 72 hpa UV: $n = 10$. (G) Experimental scheme for measuring of regenerating tentacle after UV exposure to blastema at 24 hpa (H and I). (H) Representative images of regenerating tentacles at 96 hpa. (I) Length of the regenerating tentacle after UV exposure on blastema. Ctl: $n = 12$ (tentacles), UV: $n = 12$. The numerical values that were used to generate the graphs in (C, F, and I) can be found in S1 Data. Unpaired two-tailed $t$ test. $^*p < 0.05$, $^{***}p < 0.001$. Scale bars: (H) 1 mm, (E) 100 μm, (Bi and Bii) 50 μm, (Biii and Biv) 10 μm. aPKC, atypical protein kinase C; hpa, hours post-amputation; RSPC, repair-specific proliferative cell.

*Vasa1*$^+$ cells, are primarily distributed in the bulb and branching site (Figs 3Ci, S3A–S3D, S4Ai, and S4Bi) [38,39]. By contrast, at 24 hpa *Nanos1*$^+$ cells accumulated around the injury site where blastema formed (Fig 3Cii), and *Piwi*$^+$ and *Vasa1*$^+$ cells were also localized in the blastema region (S4Aii and S4Bii Fig). While *Nanos1*$^+$ cells accumulated to the new branching bud after 48 hpa (Fig 3Ciii, white arrow) [38,39], *Nanos1* expression was no longer detected at the regenerating tip and was restricted to the middle and basal side of the tentacle (Fig 3Ciii). We further quantified the distribution of *Nanos1*$^+$ cells and found a gradual transition of their distribution from the distal to the basal side of the regenerating tentacle (Fig 3D). These results are consistent with the distribution pattern of proliferative cells (Figs 2A–2C, S2A, and S2B), further supporting the idea that proliferative cells in blastema exhibit characteristics of stem cells.

## Most blastema cells are not derived from resident stem cells localized at the bulb

Previous reports on cnidarian regeneration have suggested that undifferentiated blastema cells are supplied by the migration of resident stem/progenitor cells in response to injury [20,21,27]. To test whether resident stem cells, or originally localized i-cells, in the bulb migrate to the injury site to form blastema, we performed EdU-chase experiments during *Cladonema* tentacle regeneration. We leveraged the fact that short-time incubation of EdU labels proliferating i-cells in the intact tentacle, and, after incubating, amputated and visualized the regenerating tentacle at different time points (S5A Fig). Until 24 hpa, most EdU$^+$ cells remained localized to the bulb while a small number of EdU$^+$ cells were distributed near blastema (S5Ai–S5vi and S5B Fig). Following 48 h and 72 h chasing, EdU$^+$ cells continued to localize at the basal side, including the bulb, rather than in the regenerating tip, although the population of EdU$^+$ cells propagated in the distal direction, likely through cell proliferation and migration (S5Avii and S5Aviii Fig). Considering the minimal contribution of EdU$^+$ i-cells to blastema, these observations suggest that resident stem cells localized at the bulb may not be sufficient to form blastema.

We further conducted an alternative tracing experiment that specifically labels cells in the bulb region by DiI microinjection followed by amputation and cell tracking, which allows monitoring the labeled cells in vivo during tentacle regeneration (S6 Fig). To confirm the cell-type of DiI$^+$ cells in the bulb, we first combined EdU staining or β-catenin antibody staining with DiI injection (S6A–S6C Fig) to show that more than 75% of EdU$^+$ cells or i-cells (β-catenin cytoplasm signal) were DiI$^+$ in the bulb (S6A–S6C Fig), indicating that the majority of resident i-cells are labeled by DiI injection. After a 24 h chase, DiI$^+$ cells were still localized near the bulb, not merged with EdU$^+$ cells in blastema (S6D and S6Ei–S6ii Fig). In contrast, when DiI was injected into the central area near the amputation site, DiI$^+$ cells overlapped with EdU$^+$ cells, participating in blastema formation (S6D and S6Eiii–S6iv Fig). These results, taken together with the EdU chase results, suggest that resident stem cells do not migrate to the blastema region.

The above experiments support the presence of 2 populations of proliferative cells during regeneration: one population are the "resident homeostatic stem cells (RHSCs)," composed of cycling i-cells localized at the bulb, and the other are "repair-specific proliferative cells (RSPCs)" that appear as blastema upon amputation. In order to test this hypothesis, we performed chasing experiments by labeling the 2 proliferative cell populations with distinct nucleoside analogs, EdU and BrdU (Fig 4A). This approach has been utilized in different animal systems as a way to track proliferative cell populations and their lineages [50–52]. We labeled RHSCs in the bulb by EdU before amputation and RSPCs in the blastema by BrdU at 24 hpa (Fig 4A). At the time of blastema formation at 24 hpa, EdU$^+$ RHSCs were still localized to the bulb, while BrdU$^+$ RSPCs accumulated at the blastema (Fig 4Bi). Importantly, only 13.6% of BrdU$^+$ RSPCs were EdU$^+$ in blastema, supporting the notion that the migration of RHSCs may not be sufficient to form blastema (Fig 4Bi, 4Cii and 4D). Indeed, following 24 to 48 h, BrdU$^+$ RSPCs were distributed on the distal side of the regenerating tentacle, EdU$^+$ RHSCs remained localized on the basal side, although their distribution expanded slightly (Fig 4Bii and 4Biii). Consistent with these observations, when DiI was injected to the center area near the injury site, these DiI$^+$ cells initially contributed to RSPCs and were gradually distributed to distal side in the regenerating tentacle (S6H and S6I Fig). These results suggest that RSPCs at the blastema participate in the formation of the new structure while RHSCs expand locally, supporting the existence of 2 distinct proliferative cell populations during tentacle regeneration (Fig 4G).

To determine the characteristics of these distinct proliferative cell populations, we examined their respective proliferative capacities. For this purpose, we quantified the increase in cell numbers of the 2 populations during tentacle regeneration. At 24 hpa, RHSCs increased approximately 2-fold, but no dramatical increase was observed in the following 48 h (Fig 4E; 0 hpa: $n = 206 \pm 66.6$, 24 hpa: $n = 441.3 \pm 100.6$, 48 hpa: $n = 493.3 \pm 123.2$, 72 hpa: $n = 593.7 \pm 250.3$). Notably, the number of these RHSCs also increased during 24 h without amputation (S5C and S5D Fig), suggesting that the increase of RHSCs is not an injury-specific response. In contrast, the number of RSPCs in the blastema tripled at 48 h, which led to an approximate 4-fold increase at 72 hpa (Fig 4F; 24 hpa: $n = 275.1 \pm 64.2$, 48 hpa: $n = 929.3 \pm 151.9$, 72 hpa: $n = 1,125 \pm 225.1$). These results indicate a pronounced difference in proliferative activity between the 2 populations, further implicating their distinct responses to injury.

If RSPCs do not derive from RHSCs in the bulb, blastema should form when such stem cells are removed. To test this possibility, we examined blastema formation after RHSCs are eliminated by X-ray irradiation. We first determined a condition that would eliminate resident stem cells after exposure to different doses of X-ray irradiation (30, 50, 75 Gy) by monitoring expression of the stem cell marker *Nanos1*. FISH staining of *Nanos1* showed that *Nanos1*$^+$ cells decreased in an X-ray dose-dependent manor, and most *Nanos1*$^+$ cells were removed at 75 Gy at 3 days post-irradiation (dpi) (S7A–S7C Fig). qPCR analysis further confirmed a significant reduction of *Nanos1* mRNA levels at 3 dpi (S7G Fig). We thus amputated the tentacle at 3 dpi and investigated blastema formation by EdU pulse labeling (Fig 4H). At 24 hpa, blastema formed successfully regardless of X-ray irradiation, although the rate of EdU$^+$ cells in the blastema decreased slightly (Fig 4I and 4J). In order to support the conclusion derived from X-ray irradiation, we also performed pharmacological treatments using hydroxyurea (HU) or Mitomycin C, both of which are used to eliminate i-cells in other hydrozoan species [20,53,54]. The results of FISH and qPCR confirmed that most *Nanos1*$^+$ cells are removed by HU or Mitomycin C treatment for 3 days (S7D–S7G Fig). We then examined blastema formation after HU or Mitomycin C treatment and found that blastema formed at 24 hpa, consistent with the results of X-ray irradiation (Fig 4I and 4J). Combined, these results suggest that distinct proliferative

cell populations appear during tentacle regeneration and that most blastema cells are not derived from RHSCs.

## RHSCs contribute to nematogenesis and tentacle elongation

In the tentacles of hydrozoan medusae such as *Clytia* and *Cladonema*, proliferative i-cells in the bulb are proposed to migrate from the basal to the distal side while differentiating into nematocytes, which is known as the belt conveyor model [24,37,55]. Based on this proposed model, we hypothesized that RHSCs in the bulb contribute to nematogenesis during both homeostasis and regeneration. In order to test this hypothesis, we performed EdU and BrdU double chase experiments with some modifications (Fig 5A). We found that long-incubation with EdU for 24 h allows for the labeling of RHSCs in the bulb as well as nematocyte progenitors labeled by *Mcol1* (Figs 5Bi and S8). After EdU labeling for 24 h, followed by amputation, most EdU$^+$ cells remained in the bulb, and 26.9% of the BrdU$^+$ RSPCs were EdU$^+$ in the blastema, supporting the notion that the majority of RSPCs are of different lineage than RHSCs (Fig 5Bii, 5Biv and 5C). This result is consistent with the results of the chasing experiment using short duration EdU labeling (Fig 4A–4D), further supporting the conclusion that the migration of RHSCs in the bulb is not sufficient to form blastema. At 72 hpa, while EdU$^+$ cells were still distributed in the basal region, including the bulb, some EdU$^+$ cells were located in the distal nematocyte clusters (Fig 5Bv: yellow arrow), suggesting that one of the lineages of RHSCs are nematocytes during regeneration. We also performed DiI labeling of the bulb to show that these DiI$^+$ cells were distributed in nematocyte clusters in the regenerated tentacle (S6F and S6G Fig: white arrow).

To further confirm the above observations, we identified cell types located in the distal side of the tentacle. We found that atypical protein kinase C (aPKC) antibody can be utilized to detect cell types with nuclear staining, which is a method similar to using β-catenin antibody (Figs 3A, 5D, and S9) [39,48]. The combination of EdU chasing and cell type identification revealed that nearly half of the EdU$^+$ cells are nematocytes in the distal side of the tentacle (nematocytes: 45.7%, epithelial cells: 24.2%, i-cells: 30.1%; Fig 5E–5G). These results together indicate that RHSCs in the bulb differentiate into nematocytes during regeneration. Intriguingly, we also noticed that EdU-labeled RHSCs merged with FMRFamide$^+$ neurons (S10A and S10B Fig), suggesting that RHSCs can differentiate into neurons during tentacle regeneration.

Given that the RHSCs mainly differentiate into both nematocytes and epithelial cells during regeneration (Fig 5F and 5G), it is expected that these cells play a role in supplying new nematocytes as well as in tentacle elongation. We first examined the timing of nematocyte cluster formation as a readout of nematogenesis when RHSCs were eliminated by X-ray or drug treatments using HU or Mitomycin C (Fig 5H–5J) and found that it was delayed, particularly at the early stage of regeneration after elimination of RHSCs (Fig 5I and 5J), indicating that RHSCs migrate from the basal to the distal side while differentiating into nematocytes during regeneration.

Next, we measured tentacle length during regeneration with RHSCs cells removed by X-ray irradiation or drug treatments (Fig 5H and 5K). When tentacles were amputated at 3 dpi by which time RHSCs had been eliminated by X-ray, tentacle elongation was severely inhibited compared to the non-irradiated controls (Fig 5K, regenerated tentacle length at 4 dpa, Control: 1.66 mm, X-ray: 0.540 mm). It is possible that X-ray irradiation affects the source of RSPCs to some extent, and indeed when RHSCs were removed with HU or Mitomycin C treatment, tentacle elongation was mildly but still significantly inhibited (Fig 5K, regenerated tentacle length at 4 dpa, HU: 1.14 mm, Mitomycin C: 1.19 mm). Given that EdU$^+$ blastema cells appear after X-ray irradiation as well as drug treatments (Fig 4I and 4J), RHSCs contribute to tentacle elongation independently of blastema cells. We speculate that RHSCs, and particularly their

decedents, participate in tissue elongation likely through pushing out from the basal side as progenitors and differentiating into epithelial cells.

## RSPCs contribute to reconstruction of the new tentacle

In order to investigate the role of RSPCs during regeneration, we performed EdU chase labeling of the blastema with cell type identification by aPKC antibody staining (Fig 6A–6C). While 94.4% of the RSPCs exhibited i-cell morphology at 24 hpa (Fig 6Bi, 6Biii and 6C, consistent with the results of β-catenin antibody staining in Fig 3A and 3B), 59.2% of EdU$^+$ cells were epithelial cells in the distal portion of the regenerating tentacle at 72 hpa (Fig 6Bii, 6Biv and 6C). Strikingly, in contrast to RHSCs, EdU$^+$ cells derived from the blastema did not become nematocytes (Figs 5E–5G and 6A–6C). These results suggest that RSPCs preferentially differentiate into epithelial cells in the regenerating tentacle. Of note, we found that a small portion of EdU-labeled RSPCs merged with FMRFamide$^+$ neurons, although these populations are significantly smaller than the RHSCs (S10C–S10E Fig). Combined, these findings indicate that RSPCs are multipotent, or at least bipotent, stem cells that preferentially generate epithelial cells during regeneration.

To identify the role of RSPCs, we examined the impact of locally inhibiting cell proliferation on tentacle regeneration (Figs 6D–6I and S11). UVA radiation (320 to 400 nm) induces various forms of cellular damage via the production of reactive oxygen species or the impairment of proteasomal function, which blocks cell cycle progression at the G$_2$/M phase in both mammalian cultured cells and fly epidermal progenitors [56–59]. In addition, Hoechst and BrdU are known photosensitizers [60,61], which can facilitate the process of damage induction. To verify the effects of UVA on RSPCs, we performed UV laser illumination around the blastema region at 25 hpa and counted the number of EdU-labeled cells (Figs 6D–6F). Following around 24 to 48 h, the number of EdU$^+$ cells decreased significantly in the UV-exposed tentacle compared to the control, indicating that UV laser exposure inhibits cell proliferation of RSPCs (Fig 6F). In contrast, we confirmed that cell proliferation and the timing of nematocyte cluster formation by RHSCs were similar to the non-UV exposure controls, suggesting a minimal impact of UV laser exposure on RHSCs (S11 Fig). In order to examine the contribution of the proliferation of RSPCs toward tentacle regeneration, we monitored the length of regenerating tentacles after UV laser exposure (Fig 6G). Upon UV laser illumination, tentacle elongation during regeneration was significantly inhibited (Fig 6H and 6I). These results, together with the distribution of their descendants on the distal side of the regenerating tentacle (Figs 4Biii and 5Biii), indicate that proliferation of RSPCs contribute to the reconstruction of new tentacles after amputation.

## Discussion

In this study, we uncovered the cellular mechanism underlying blastema formation during *Cladonema* tentacle regeneration. We found that, upon amputation of the tentacle, RSPCs accumulate as blastema, most of which is not derived from RHSCs. While RHSCs contribute to nematogenesis and tissue elongation during both homeostasis and regeneration, RSPCs differentiate into the epithelial cells of the newly formed tentacle. Based on these results, we propose that distinct proliferative cell populations facilitate rapid functional regeneration of the tentacle, which enables critical behaviors of jellyfish organismal physiology such as feeding and swimming (Fig 7A).

## Heterogeneity of stem-like cells

In hydrozoans, i-cells behave as pluripotent or multipotent stem cells that can differentiate into several progenitors and differentiated lineages such as nematocytes, neurons, gland cells,

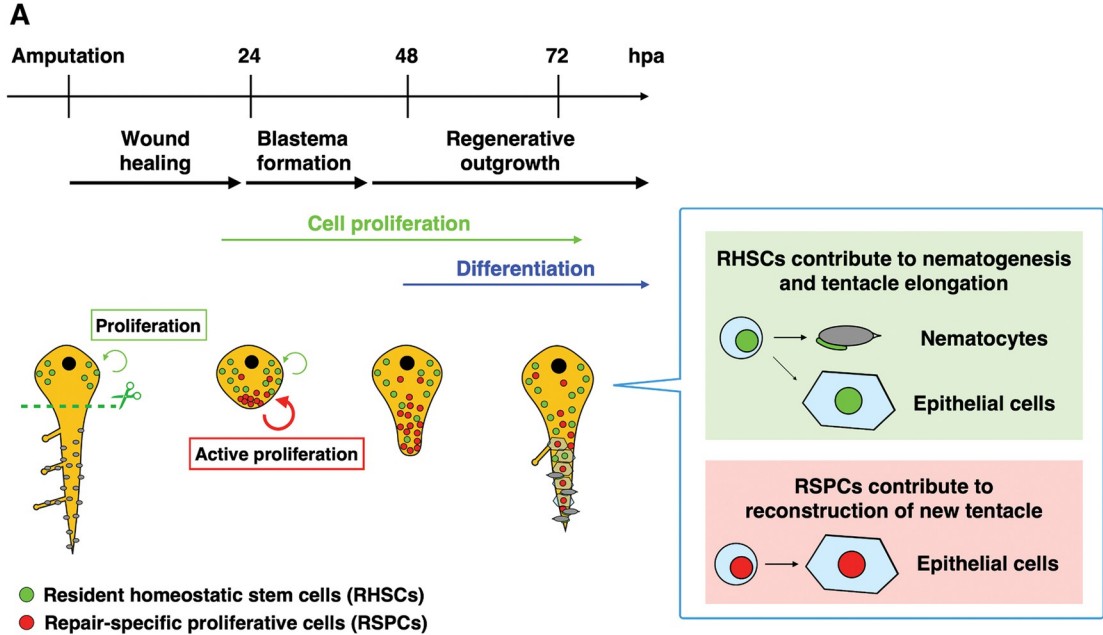

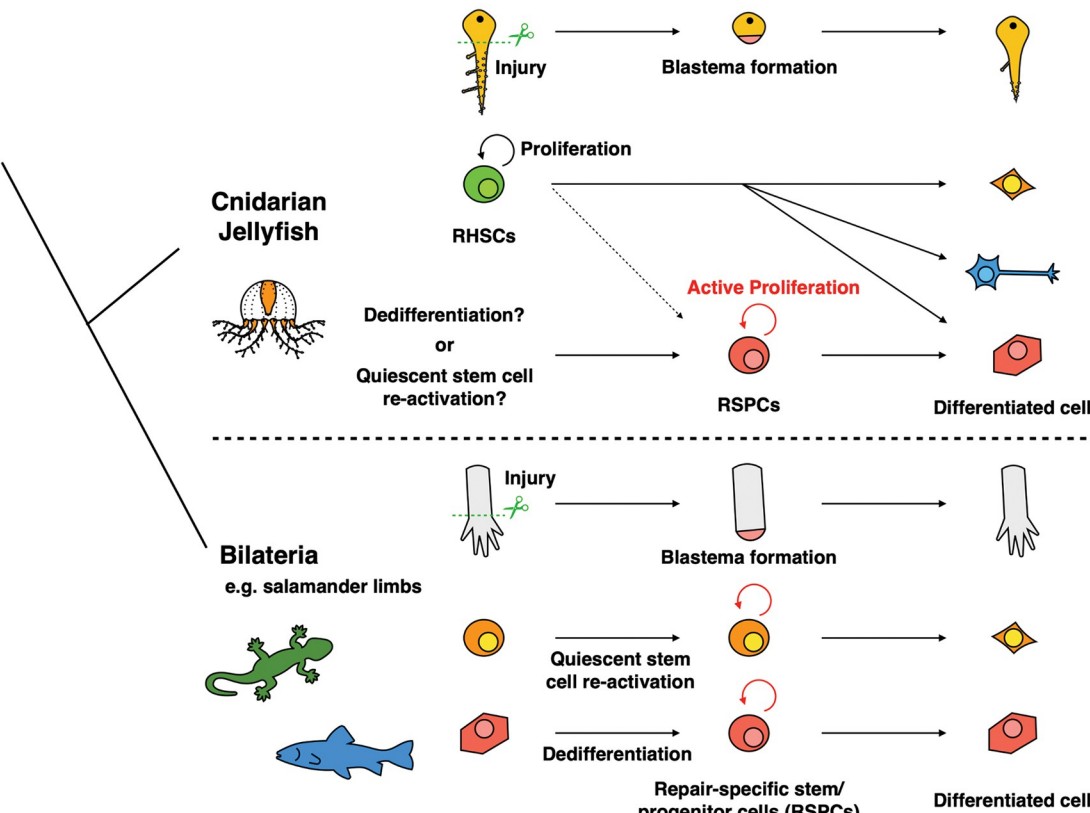

**Fig 7. Summary of *Cladonema* tentacle regeneration and evolutionary history of appendage regeneration.** (A) When a *Cladonema* medusa tentacle is amputated, wound closure is completed by 24 hpa. After wound healing, RSPCs appear and actively proliferate at the wound tip. Ectopic cell proliferation on the distal side is gradually suppressed, and the cell differentiation phase begins. While RHSCs in the bulb continuously proliferate and supply nematocytes and epithelial cells, RSPCs mainly differentiate into epithelial cells. (B) Blastema formation mechanism during appendage regeneration across animal evolution. When an

appendage is amputated, RSPCs (stem/progenitor cells) contribute to blastema formation in both regenerative bilaterians (e.g., salamanders, zebrafish, and crickets) and cnidarian jellyfish. hpa, hours post-amputation; RHSC, resident homeostatic stem cell; RSPC, repair-specific proliferative cell.

and gametes. Although i-cells exhibit common features such as their interstitial localization, cellular morphology, and expression of the conserved stem cell markers (*Nanos*, *Piwi*, *Vasa*, *PL10*); their potency and heterogeneity likely differ across species [45]. Indeed, *Hydra* i-cells differentiate into most cell types except epithelial cells, while *Hydractinia* i-cells behave as pluripotent cells that can differentiate into all cell types including somatic and germline cells [16,62,63].

In the intact *Cladonema* jellyfish, $Nanos1^+$, $Piwi^+$, and $Vasa1^+$ cells are primarily distributed at the bulb (Figs 3C, S4A, and S4B), and about 70% of RHSCs in the bulb are $Nanos1^+$ cells (Fig 3E and 3F). From nucleoside pulse-chase experiments, these RHSCs differentiate into epithelia as well as nematocytes (Fig 5F and 5G), suggesting their role as multipotent stem cells. These observations are consistent with the jellyfish *Clytia*, in which i-cells in the bulb proliferate and differentiate into nematocytes [37,55], indicating that i-cells in the tentacle bulb are commonly involved in organ homeostasis including nematogenesis in hydrozoan jellyfish.

In contrast to RHSCs in the bulb, only 17% to 36% of RSPCs at blastema express $Nanos1^+$, $Piwi^+$, and $Vasa1^+$ (Figs 3E, 3F, and S4C–S4E), although more than 80% of blastema cells show i-cell morphology (Figs 3A, 3B, 6B, and 6C). These results imply that the stem-like population of the blastema is likely heterogenous and differs from that of resident i-cells in the bulb. Indeed, RSPCs predominantly differentiate into epithelial cells rather than nematocytes (Fig 6A–6C), suggesting their role as lineage-restricted stem/progenitor cells while maintaining the potential to become neurons in the regenerating tentacle (S10 Fig). Such stem-like cells also accumulate at the newly formed branch starting at 48 to 72 hpa (Fig 4Bii–4iii), consistent with the prediction by the previous report proposing that i-cell population in the branching bud differs from RHSCs in the bulb [39]. While previous studies using other cnidarians have described active cell proliferation upon injuries, these proliferating cells are pluri/multipotent stem cells migrating to the injury site [20,21,27,64], not lineage-restricted stem/progenitor cells like RSPCs. Altogether, our findings support the heterogeneity of stem-like cells in *Cladonema* jellyfish (Fig 7A), which is similar to the subpopulations of *Hydra* i-cells [54,65].

## How are RSPCs supplied in the blastema?

Our work is the first to provide evidence that RSPCs, which exhibit preferential differentiation into epithelial cells, appear upon injury in cnidarians, but how this cell population is supplied remains unknown. Nucleoside pulse-chase experiments, together with the elimination of RHSCs, show that blastema formation is mostly independent of RHSCs in the bulb (Figs 4 and S6). When the distal side of the tentacle, where few proliferating cells exist, is amputated, local cell proliferation is still observed around the injury site (Fig 2G). These observations raise 2 possibilities: mitotic re-entry of quiescent/slow-cycling stem cells or dedifferentiation from differentiated cells into stem-like cells.

Previous reports have shown that some cnidarian species possess quiescent or slow-cycling stem cells. For example, while *Hydra* has 3 types of adult stem cells—i-cells, epidermal stem cells, and gastrodermis epithelial stem cells—the subpopulation of each stem cell is slow-cycling [54]. These slow-cycling cells are resistant to hydroxyurea treatment due to G2 phase arrest and re-enter the cell cycle during head regeneration. *Nematostella* also has slow-cycling/quiescent cells that are predominantly present in the mesenteries and are resistant to X-ray irradiation, and these cells' cell cycle is activated upon injury, when they migrate toward the

wound site [21]. Given that a small number of i-cells exist in the *Cladonema* intact tentacle, outside the bulb, and branching bud (Figs 3C, S3A, S3C, S3D, S4A and S4B) [39]; such a minor population of i-cells may behave as slow-cycling/quiescent cells and re-enter the cell cycle in response to injury.

By contrast, although cnidarian dedifferentiation potential is still debated, some species appear to exhibit context-dependent cellular plasticity [66]. For example, classical experiments using the hydrozoan jellyfish *Podocoryna carnea* have indicated their dedifferentiation potential in vitro: when isolated striated muscles are cultured after collagenase treatment, the muscle cells dedifferentiate into stem-like cells and re-differentiate into multiple cell types [67]. Additionally, in green *Hydra*, the isolated gastrodermis, which is i-cell free, can reconstitute a complete polyp including the epidermis [68]. In this case, gastrodermal gland cells appear to dedifferentiate into i-cells [69], while gastrodermal digestive cells transdifferentiate into epidermal epitheliomuscular cells [70]. Moreover, a recent report using *Hydractinia* has shown that, when the hypostome is isolated from the body, cellular reprogramming triggered by cell senescence occurs in somatic cells, which convert into pluripotent *Piwi1*$^+$ i-cells [71]. Although these reports support the possibility that dedifferentiation produces pluri/multipotent stem cells after isolation from the body, it is also possible that RSPCs are derived from their descendant cell types. During newt limb regeneration, cellular lineage is restricted such that muscle progenitors dedifferentiated from resident muscle fibers produce only muscles [7,72]. Given that RSPCs in the *Cladonema* tentacle mainly differentiate into epithelial cells (Fig 6A–6C), epithelial cells may dedifferentiate into progenitors that give rise to epithelial cells in the newly formed tentacle.

## Evolutionary conservation and diversification of blastema formation

Although the molecular and cellular mechanisms of animal regeneration have been addressed in detail using representative species such as planaria and salamanders, questions such as the evolutionary history of regeneration and the homology of regeneration mechanisms across species remain largely unknown. Repair-specific stem cells, or RSPCs, provided by quiescent stem cells and/or dedifferentiation are common blastema components in bilaterian appendage regeneration [3,73]. As in vertebrates where lineage-restricted repair-specific stem/progenitor cells participate in appendage regeneration [7,9,11], the protostome ecdysozoa *Parhyale hawaiensis* utilize re-activation of muscle satellite cells during thoracic leg regeneration [74]. By contrast, during echinoderm starfish arm regeneration, dedifferentiated dermal cells form blastema-like structures [75]. Similarly, during insect cricket leg regeneration, the source of blastemal cells is derived from injury-induced dedifferentiation [76,77]. These reports suggest that repair-specific production of blastema cells is relatively conserved throughout bilaterians (Fig 7B).

In non-bilaterian metazoans, resident pluripotent stem cells are thought to contribute to both homeostasis and whole-body regeneration. During the regeneration of the demosponge in Porifera, one of the most primitive animals, archeocytes known as homeostatic stem cells contribute to an undifferentiated cell mass beneath the wound as a blastema-like structure [78]. While the regenerating ctenophore cydippids do not apparently form blastema-like structure, slow-cycling stem-like cells are recruited to the injury site [79]. In cnidarians, resident stem cells, which are involved in homeostasis, are recruited to the injury site after amputation of the body and removal of the organs [19–21,27]. By contrast, as shown in this study, the main cellular source of blastemal cells is not RHSCs but rather RSPCs during *Cladonema* tentacle regeneration (Fig 4). Our findings suggest that, in terms of the regeneration processes and blastemal cellular source, the regeneration mechanism of the *Cladonema* tentacle is more similar to that of bilaterian species (Fig 7B).

One important remaining question is whether bilaterians' and cnidarians' most recent common ancestor possessed a system to supply blastema through repair-specific stem/proliferative cells. Although sponges and ctenophores have been proposed to possess dedifferentiation or transdifferentiation potentials that can provide a cellular source of regenerated bodies [66], it is currently unknown whether they are repair-specific cell populations or not. In this study, we identify RSPCs that contribute to the newly generated epithelium during *Cladonema* tentacle regeneration. Intriguingly, recent genomic studies have revealed that jellyfish-specific genes are not conserved among bilaterians or even other cnidarians including corals and sea anemones, and the anthozoan *Nematostella* is the most "bilaterian-like" cnidarian sequenced [80–82]. These findings raise the possibility that medusozoans had evolved the medusa stage after branching from their common ancestor and that bilaterians and cnidarian jellyfish likely independently acquired a similar mechanism of blastema formation upon amputation of appendages. It is thus tempting to speculate that blastema formation by RSPCs is a common feature acquired for complex organ and appendage regeneration during animal evolution.

## Limitations, alternative interpretations, and future directions

Given the current lack of knowledge concerning the exact cell cycle and the inherent heterogeneity of *Cladonema* i-cells, it is plausible that a subset of i-cells located at the bulb, not marked by 24 h EdU uptake, may become activated after amputation and contribute to blastema formation. Furthermore, the cellular origins of RSPCs in the blastema remain elusive, with at least 2 conceivable scenarios: mitotic re-entry of quiescent stem cells and/or dedifferentiation of neighboring specialized cells. Due to the current constraints imposed by available tools and techniques, addressing these issues is challenging at this stage.

Looking forward, it is essential to introduce genetic tools that allow the tracing of specific cell lineages and the manipulation of genes in *Cladonema*. This approach promises to deepen our understanding of jellyfish development and regeneration at the molecular and cellular levels and also to enhance the utility of *Cladonema* as a more useful research model.

## Supporting information

**S1 Fig. Wound healing, neuron regeneration, and the rate of regeneration.** (A) Representative images of nematocyte clusters (NCs) in the medusa tentacle. Yellow arrows indicate nematocyte clusters that include mature nematocytes (Poly-γ-glutamate$^+$). (B) The process of wound healing from 4 hpa to 24 hpa with Phalloidin and anti-α-Tubulin antibody. Phalloidin for F-actin (red) and α-Tubulin (green). (C) The extent of wound healing during tentacle regeneration. Full open: actin fibers not attached (Bi), partially close: disorganized actin fibers (Bii and Biii), completed close: actin fibers are fully attached (Biv); 4 hpa: *n* = 15 (tentacles), 8 hpa: *n* = 17, 16 hpa: *n* = 15, 24 hpa: *n* = 14. (D) Neural morphology in intact tentacle and the regenerating tentacle stained with the anti-FMRFamide antibody. Yellow allows indicate cell bodies of FMRFamide neurons. FMRFamide (magenta). (i, ii) Intact whole tentacle, (iii, iv) basal side of intact tentacle, (v, vi) regenerating tentacle at 24 hpa. (E) Neural morphology in the regenerating tentacle stained with the anti-acetylated-Tubulin antibody. White arrows indicate neural fibers; acetylated-Tubulin (green). (F) The rate of the tentacle regeneration after removing the bulb from canals. Difference of the regeneration rate between each experiment. (G) Images of the isolated tentacle without bulb at 1 dpa and 7 dpa. (H) Images of the isolated whole tentacle 1 dpa and 7 dpa. The numerical values that were used to generate the graphs in (**C and F**) can be found in S1 Data. Scale bars: (A, Di, Diii, Dv, Ei, and Eiii) 100 μm, (Div, Dvi, Eii, and Eiv) 50 μm, (G and H) 1 mm.
(TIFF)

**S2 Fig. The distribution of mitotic cells during regeneration.** (A) Distribution of mitotic cells detected by anti-PH3 in regenerating tentacle. (B) Quantification of PH3$^+$ cells' distribution in regenerating tentacle based on the defined areas in (A). Intact: $n = 11$ (tentacles), 24 hpa: $n = 10$, 48 hpa: $n = 9$, 72 hpa: $n = 8$. (C) Distribution of S-phase cells in regenerating tentacle stained with EdU 1 h pulse labeling. EdU 1 h pulse labeling was performed at 0, 4, 8, 12, 16, 20, and 24 hpa as shown in the scheme. White dot squares show each quantification area ($150^2$ $\mu m^2$) in (D and E). (D) EdU relative intensity in blastema; 0 hpa: $n = 6$, 4 hpa: $n = 7$, 8 hpa: $n = 10$, 12 hpa: $n = 11$, 16 hpa: $n = 10$, 20 hpa: $n = 26$, 24 hpa: $n = 12$. (E) The number of EdU$^+$ cells in blastema; 0 hpa: $n = 6$, 4 hpa: $n = 7$, 8 hpa: $n = 10$, 12 hpa: $n = 11$, 16 hpa: $n = 10$, 20 hpa: $n = 26$, 24 hpa: $n = 12$. $^*p < 0.05$, $^{***}p < 0.001$. The numerical values that were used to generate the graphs in (**B, D, and E**) can be found in S1 Data. Scale bars: (A and C) 100 $\mu m$. (TIFF)

**S3 Fig. The distribution of stem cells in intact tentacle.** (A–D) Expression of *Nanos1*, *Nanos2*, *Piwi*, and *Vasa1* in intact tentacles by FISH. Yellow arrows indicate the branching site. (E) Expression of *Nanos1* and *Nanos2* in intact tentacle by double FISH. White arrowheads indicate *Nanos1*$^+$, and white arrows indicate *Nanos2*$^+$. (F) Expression of *Nanos2* and *Mcol1* in intact tentacle by double FISH. Yellow arrows indicate representative co-expression of *Nanos2* and *Mcol1*. Scale bars: (A–D) 100 $\mu m$, (Ei and Fi) 50 $\mu m$, (Eii and Fii) 25 $\mu m$. (TIFF)

**S4 Fig. The distribution of stem cells in regenerating tentacle.** (A and B) Distribution of cells with stem marker genes (*Piwi*$^+$ or *Vasa1*$^+$) in intact tentacle and regenerating tentacle by FISH. Yellow arrow indicates the branching site. (C and D) Images of the regenerating tentacle co-labeled with *Piwi* or *Vasa1* FISH and EdU 1 h pulse labeling at 24 hpa. EdU (green) and *Piwi* or *Vasa1* (magenta). White squares show each quantification area ($74.72^2$ $\mu m^2$) in (E). Yellow arrowheads indicate EdU$^+$/*Piwi*$^+$ or EdU$^+$/*Vasa1*$^+$ cell. (E) The rate of cells positive for stem cell marker genes in blastema. *Nanos1*: $n = 22$, *Piwi*: $n = 12$, *Vasa1*: $n = 11$. The data of "*Nanos1*" are the same as that in Fig 3F (*Nanos1*+/EdU+ cells in blastema). The numerical values that were used to generate the graphs in (**E**) can be found in S1 Data. Scale bars: (Aiv and Biv) 100 $\mu m$, (Ai-iii, Bi-iii, Ci, and Di) 50 $\mu m$, (Cii and Dii) 25 $\mu m$. (TIFF)

**S5 Fig. Distribution and increase of resident homeostatic stem cells (RHSCs) during tentacle regeneration.** (A) Transition of labeled RHSCs during regeneration by chasing EdU$^+$ cells. EdU 1 h labeling before amputation and chasing at 0, 4, 8, 12, 16, 24, 48, and 72 hpa. White dot squares show each quantification area ($150^2$ $\mu m^2$) in (B). (B) The number of EdU$^+$ cells around blastema during regeneration; 0 hpa: $n = 6$, 4 hpa: $n = 6$, 8 hpa: $n = 13$, 12 hpa: $n = 12$, 16 hpa: $n = 8$, 24 hpa: $n = 10$. (C) The comparison of proliferative cell number in intact vs. no amputation vs. 24 hpa. EdU 1 h pulse labeling before amputation and chasing with amputation or without amputation (no amp). (D) The number of EdU$^+$ cells labeled before amputation. Counted area is the whole regenerating tentacle at 24 hpa. Detailed information is in (E). Intact: $n = 6$ (tentacles), 24 h (no amp): $n = 6$, 24 hpa: $n = 6$. (E) Area size used for quantification. Note that the area of quantification at 24 hpa is similar. The numerical values that were used to generate the graphs in (**B, D, and E**) can be found in S1 Data. Unpaired two-tailed $t$ test. $^{***}p < 0.001$. Scale bars: (A and C) 100 $\mu m$. (TIFF)

**S6 Fig. DiI injection revealed that resident stem cells do not migrate to blastema during regeneration.** (A) Experimental scheme of DiI-labeled cell identification in the bulb with EdU

or anti-β-catenin staining in (B). (B) Co-staining of DiI and EdU or anti-β-catenin in intact tentacle. White dot squares show each quantification area ($74.72^2$ μm$^2$) in (C). White arrowheads indicate DiI$^+$/EdU$^+$ and white arrows indicate EdU$^+$ only. Yellow arrowheads indicate DiI$^+$/i-cells (β-catenin cytoplasmic signal$^+$ cells) and yellow arrows indicate DiI$^-$/i-cells. (C) Rate of DiI-labeled cells per proliferative cells or i-cells in bulb. Quantification of DiI$^+$/EdU$^+$ cells or DiI$^+$/i-cells. DiI$^+$/EdU$^+$ cells: $n = 5$ (areas), DiI$^+$/i-cells: $n = 5$. (D) Experimental scheme of chasing DiI-labeled cells in (E). EdU 1 h pulse labeling at 24 hpa. (E) (i and ii) Little migration of DiI-labeled cells in bulb from the moment of amputation to 24 hpa. $n = 9/9$. (iii and iv) DiI-labeled cells in center area that merged with EdU (yellow arrow). $n = 10/12$. (F) Experimental scheme for chasing DiI-labeled cells in the bulb in (G). (G) Distribution change of DiI-labeled cells in the bulb during tentacle regeneration of an animal. White arrow indicates migration of labeled cells to nematocyte cluster at distal side. $n = 6/10$. (H) Experimental scheme for chasing DiI-labeled cells in the central area of tentacle in (I). (I) Distribution change of DiI-labeled cells in central area during tentacle regeneration of an animal. White arrow indicates DiI-labeled cells in the newly regenerated tentacle. $n = 17/19$. The numerical values that were used to generate the graphs in (C) can be found in S1 Data. Scale bars: (G and I) 250 μm, (Bi, Biii, and E) 100 μm, (Bii and Biv) 50 μm.
(TIFF)

**S7 Fig. *Nanos1*$^+$ cells in the bulb decrease after X-ray irradiation or drug treatments.** (A) Experimental scheme of X-ray irradiation (30, 50, and 75 Gy). FISH and qPCR at 3 days post-irradiation (3 dpi). (B) Expression of *Nanos1* in intact tentacle at 3 dpi by FISH. (C) Quantification of *Nanos1*$^+$ cells number in tentacle. Quantification area is the entire tentacle in confocal images. Control: $n = 9$, 30 Gy: $n = 5$, 50 Gy: $n = 5$, 75 Gy: $n = 17$. (D) Experimental scheme of drug treatments (HU 10 mM or Mitomycin C 30 μm). FISH and qPCR after 3 days of treatment. (E) Expression of *Nanos1* in intact tentacle after 3 days drug treatments by FISH. (F) Quantification of *Nanos1*$^+$ cells in tentacle. Quantification area is the entire tentacle in confocal images. Control: $n = 9$, 75 Gy: $n = 17$, HU: $n = 10$, Mitomycin C: $n = 8$. The data of "Control" and "X-ray 75 Gy" are the same as that in S7C Fig (Control and 75 Gy, respectively). (G) Relative expression of *Nanos1* after irradiation and drug treatments by qPCR. The numerical values that were used to generate the graphs in (**C, F, and G**) can be found in S1 Data. Unpaired two-tailed $t$ test. *$p < 0.05$, ***$p < 0.001$. Scale bars: (B and E) 50 μm.
(TIFF)

**S8 Fig. EdU long incorporation labels nematocyte progenitors.** (A) Nematocyte progenitor cells in the bulb with dual staining of *Mcol1* FISH and EdU staining. EdU 1 h or 24 h pulse labeling. The expression level of *Mcol1* is extremely high such that the *Mcol1* signal invades the wavelength of the EdU signal. White arrows indicate EdU$^+$ cells and yellow arrowheads indicate *Mcol1*$^+$/EdU$^+$ cells. (B) The rate of *Mcol1*/EdU$^+$ cells in intact tentacle. Quantification area is the entire tentacle in single section of confocal images. 1 h: $n = 8$ (tentacles), 24 h: $n = 8$. The numerical values that were used to generate the graphs in (**B**) can be found in S1 Data. Unpaired two-tailed $t$ test. **$p < 0.005$. Scale bars: (A) 50 μm.
(TIFF)

**S9 Fig. Similar pattern between β-catenin and aPKC antibody staining.** Identification of cell types in blastema stained by β-catenin and aPKC antibody staining. Scale bar: 50 μm.
(TIFF)

**S10 Fig. Neurons are newly generated from 2 stem-like cell populations during regeneration.** (A) Experimental scheme of chasing RHSCs in (B). (B) Neuron derived from RHSCs by co-staining with EdU and anti-FMRFamide antibody at 72 hpa. Yellow arrowhead indicates

EdU$^+$ and FMRFamide$^+$ cell. (C) Experimental scheme of chasing RSPCs in (D). (D) Neuron derived from RSPCs by co-staining with EdU and anti-FMRFamide antibody at 72 hpa. Yellow arrowhead indicates EdU$^+$ and FMRFamide$^+$ cell. (E) (i) The number of neurons in each chasing experiment. Quantification area is the entire tentacle in confocal images (Bii and Dii). RSPCs: $n = 19$ (images), RHSCs: $n = 18$. (ii) Rate of EdU$^+$/FMRFamide$^+$ cell in (Ei). The numerical values that were used to generate the graphs in (**E**) can be found in S1 Data. Unpaired two-tailed $t$ test. $^*p < 0.05$. Scale bars: (Bi and Di) 100 μm, (Bii and Dii) 50 μm. (TIFF)

**S11 Fig. Blastema exposure to UV has no significant effect on RHSCs.** (A) Experimental scheme depicting the combination of EdU chasing and UV exposure in (B–D). (B) Bleaching of Hoechst signal only in UV exposure area at 24 hpa. White dot square is UV exposure area. (C) Distribution of EdU-labeled cells from intact to 72 hpa, Ctl (no UV) vs. UV. (D) The number of EdU$^+$ cells during regeneration, Ctl (no UV) vs. UV. Intact: $n = 13$ (tentacles), 24 hpa: $n = 9$, 48 hpa Ctl: $n = 11$, 48 hpa UV: $n = 7$, 72 hpa Ctl: $n = 9$, 72 hpa UV: $n = 7$. (E) Experimental scheme depicting nematogenesis monitoring after UV exposure in (F and G). (F) Representative images of Ctl vs. UV at 72 hpa. Yellow arrowheads show nematocyte clusters. (G) Timing of nematocyte cluster formation during tentacle regeneration, Ctl vs. UV. Each tentacle: $n = 12$. The numerical values that were used to generate the graphs in (**D and G**) can be found in S1 Data. Unpaired two-tailed $t$ test. $^{***}p < 0.001$. Scale bars: (F) 500 μm, (B and C) 100 μm. (TIFF)

**S1 Data. Numerical values for all main and supporting figures.** (XLSX)

## Acknowledgments

We thank R. Deguchi (Miyagi Univ. Education, Japan) for sharing *Cladonema pacificum*. We thank D. Umetsu, H. Nagai, and T. Akiyama for helpful discussion. We thank I. Nagai, A. Sasaki, and H. Nakatani for technical assistance and animal maintenance. We thank Y. Nashimoto for technical support.

## Author Contributions

**Conceptualization:** Sosuke Fujita, Yu-ichiro Nakajima.

**Data curation:** Sosuke Fujita, Mako Takahashi, Yu-ichiro Nakajima.

**Funding acquisition:** Sosuke Fujita, Gaku Kumano, Erina Kuranaga, Masayuki Miura, Yu-ichiro Nakajima.

**Methodology:** Mako Takahashi, Gaku Kumano.

**Resources:** Yu-ichiro Nakajima.

**Supervision:** Gaku Kumano, Erina Kuranaga, Masayuki Miura, Yu-ichiro Nakajima.

**Validation:** Yu-ichiro Nakajima.

**Writing – original draft:** Sosuke Fujita, Yu-ichiro Nakajima.

**Writing – review & editing:** Sosuke Fujita, Gaku Kumano, Erina Kuranaga, Masayuki Miura, Yu-ichiro Nakajima.

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
