## [Decision Letter · Decision Letter 0]

9 Oct 2023

Dear Dr Nakajima,

Thank you for your patience while we considered your revised manuscript "Distinct stem-like cell populations facilitate functional regeneration of the Cladonema medusa tentacle" for publication as a Research Article at PLOS Biology. This revised version of your manuscript has been evaluated by the PLOS Biology editors, the Academic Editor and the original reviewer 1.

Reviewer 1 has commented that the manuscript has been strengthened in this revision, but s/he feels that the model of tentacle regeneration is not 'definitive'. After discussing this point with our Academic Editor, we think there is consilience around the current interpretations and we would not require further experimentation to address this point. However, in light of this reviewer comment, we think that some of the conclusions should be toned down and that you should add a section to the discussion detailing limitations, alternative interpretations, and future directions. As an example of a toned down conclusion, the Academic Editor has suggested that on page 19: "Combined, these results demonstrate that distinct proliferative cell populations appear during tentacle regeneration and that most blastema cells are not derived from RHSCs" could be changed to address this point by replacing 'demonstrate' with 'suggest'.

Based on the reviews and on our Academic Editor's assessment of your revision, we are likely to accept this manuscript for publication, provided you satisfactorily address the remaining points. Please also make sure to address the following data and other policy-related requests.

**IMPORTANT: Please address the following editorial requests: 

1) FINANCIAL DISCLOSURES: In the relevant section of our online system, please update the financial disclosures statement to describe the role of any sponsors or funders in the study design, data collection and analysis, decision to publish, or preparation of the manuscript. If the funders had no role in any of the above, include this sentence at the end of your statement: "The funders had no role in study design, data collection and analysis, decision to publish, or preparation of the manuscript."

2) BLURB: In the relevant section of our online system, please provide a blurb which (if accepted) will be included in our weekly and monthly Electronic Table of Contents, sent out to readers of PLOS Biology, and may be used to promote your article in social media. The blurb should be about 30-40 words long and is subject to editorial changes. It should, without exaggeration, entice people to read your manuscript. It should not be redundant with the title and should not contain acronyms or abbreviations. 

3) DATA AVAILABILITY: You may be aware of the PLOS Data Policy, which requires that all data be made available without restriction: http://journals.plos.org/plosbiology/s/data-availability. For more information, please also see this editorial: http://dx.doi.org/10.1371/journal.pbio.1001797

a. Supplementary files (e.g., excel). Please ensure that all data files are uploaded as 'Supporting Information' and are invariably referred to (in the manuscript, figure legends, and the Description field when uploading your files) using the following format verbatim: S1 Data, S2 Data, etc. Multiple panels of a single or even several figures can be included as multiple sheets in one excel file that is saved using exactly the following convention: S1_Data.xlsx (using an underscore).

b. Deposition in a publicly available repository. Please also provide the accession code or a reviewer link so that we may view your data before publication. 

Fig 1I; Fig 2B,E-F; Fig3B,D,F; Fig4,F,J; FIg5C,G,J0K; Fig 6C,F,I;

Fig S1C,F; Fig S2B,D,E; Fig S4E; Fig S5B,D; Fig S7C,F-G; Fig S8B; Fig S10E; Fig S11D,G;

>>Please also ensure that figure legends in your manuscript include information on where the underlying data can be found, and ensure your supplemental data file/s has a legend.

>>Please ensure that your Data Statement in the submission system accurately describes where your data can be found.

4) CODE: Per journal policy, if any code was generated to support the conclusions of your manuscript, we require that you make it available without restrictions upon publication. Please ensure that any code is sufficiently well documented and reusable, and that your Data Statement in the Editorial Manager submission system accurately describes where your code can be found.

We expect to receive your revised manuscript within two weeks. 

*Published Peer Review History*

*Press*

Sincerely,

Luke

Lucas Smith, Ph.D.

Senior Editor,

lsmith@plos.org,

PLOS Biology

Reviewer remarks:

Reviewer #1: The authors have responded to my and the other reviewers' points from the first review cycle and revised the manuscript by performing new experiments, resulting in new or improved figures. They have also edited the text at places.

The new data, in particular the 24 h EdU treatment that was followed by amputation and BrdU labeling, may indeed suggest that not all proliferative cells in the blastema directly derive from resident tentacle bulb i-cells. Therefore, it is possible that a distinct population (or populations) of cells also contributes to regeneration alongside i-cells and their progeny. This, however, isn't conclusive since the cell cycle length of Cladonema i-cells is yet unknown. A long i-cell cell cycle could still leave many i-cells unlabeled following a 24 h EdU treatment. These cells could increase their proliferation activity following amputation. Some of the other new experiments in the revised manuscript also make a better case for the authors' preferred conclusion; however, the work overall does not provide a definitive model re the cellular composition of the blastema. 

Assuming that there are indeed repair-specific cells in the blastema that are distinct from i-cells, the main question arising from this work remains their origin. The authors suggest two alternatives, one being a population of quiescent stem cells (would they call them i-cells too?), and the other is dedifferentiation of existing somatic cells. 

Minor comments:

The authors should avoid the term "lineage trace". This term refers to tracking single cells or at least a defined population that, e.g., expresses a marker gene (i.e., a lineage). What the authors performed was S-phase trace, looking at a broad group of cells that are in S-phase, which could encompass multiple lineages.

What is the evidence that cytoplasmic b-catenin marks all i-cells? Identifying i-cells by morphology may overestimate their numbers because they include committed i-cell progeny that still cycle (e.g., nematoblasts).

Related to the above point, the authors should use the term i-cell consistently. First, they have to define them in Cladonema. As it stands now, it is not always clear to which cells the authors refer. Reading the manuscript, one gets the impression that the authors themselves aren't sure. Are i-cells merely small, proliferative cells with undifferentiated morphology, expressing Nanos/Piwi/Vasa, and being able to differentiate into some (which ones?) cell types? Alternatively, if they are unsure of how to define i-cells, they should clearly state it. This issue is always hard to resolve in a new animal model.

---

## [Decision Letter · Decision Letter 1]

16 Nov 2023

Dear Dr Nakajima,

Thank you for the submission of your revised Research Article "Distinct stem-like cell populations facilitate functional regeneration of the Cladonema medusa tentacle" for publication in PLOS Biology, and apologies again for our delay in sending you a decision. As mentioned over email, your revised study was assessed by Reviewer 1, who is fully satisfied by the changes made. Therefore, on behalf of my colleagues and the Academic Editor, Kimberly L Cooper, I am pleased to say that we can in principle accept your manuscript for publication, provided you address any remaining formatting and reporting issues. These will be detailed in an email you should receive within 2-3 business days from our colleagues in the journal operations team; no action is required from you until then. Please note that we will not be able to formally accept your manuscript and schedule it for publication until you have completed any requested changes.

PRESS

Sincerely, 

Lucas Smith, Ph.D.

Senior Editor

PLOS Biology

lsmith@plos.org